# Age-related changes in diffuse optical tomography sensitivity profiles in infancy

**Xiaoxue Fu** [iD] *, **John E. Richards**

Department of Psychology, University of South Carolina, Columbia, United States of America

* xiaoxuef@mailbox.sc.edu

**Data Availability Statement:** The age-specific average templates, including T1, DOT sensitivity average template can be accessed from the "Neurodevelopmental MRI Database". The

## Abstract

Diffuse optical tomography uses near-infrared light spectroscopy to measure changes in cerebral hemoglobin concentration. Anatomical interpretations of the location that generates the hemodynamic signal requires accurate descriptions of diffuse optical tomography sensitivity to the underlying cortical structures. Such information is limited for pediatric populations because they undergo rapid head and brain development. The present study used photon propagation simulation methods to examine diffuse optical tomography sensitivity profiles in realistic head models among infants ranging from 2 weeks to 24 months with narrow age bins, children (4 and 12 years) and adults (20 to 24 years). The sensitivity profiles changed systematically with the source-detector separation distance. The peak of the sensitivity function in the head was largest at the smallest separation distance and decreased as separation distance increased. The fluence value dissipated more quickly with sampling depth at the shorter source-detector separations than the longer separation distances. There were age-related differences in the shape and variance of sensitivity profiles across a wide range of source-detector separation distances. Our findings have important implications in the design of sensor placement and diffuse optical tomography image reconstruction in (functional) near-infrared light spectroscopy research. Age-appropriate realistic head models should be used to provide anatomical guidance for standalone near-infrared light spectroscopy data in infants.

## Introduction

Diffuse optical tomography (DOT) uses near-infrared light spectroscopy (NIRS) to measure changes in cerebral hemoglobin concentration [1]. Multi-channel NIRS instruments measures hemoglobin changes in the scalp channel space. DOT instruments use overlapping and channels with multiple separation distances to enhance spatial resolution [2]. However, both types of the scalp-based measurement do not directly provide anatomical information about the brain regions of the hemodynamic signal. Localizing the channel-wise signals to specific brain regions requires comprehensive understanding of the DOT sensitivity profile. It is difficult to experimentally measure DOT sensitivity inside human head [3]. Instead, Monte Carlo simulations are used to simulate the propagation of photons through head and brain tissues [4]. The forward model can then guide DOT image reconstruction to recover the brain locations of hemoglobin concentration changes [5, 6]. DOT sensitivity profiles have been studied in adults.

Database is available online: http://jerlab.psych.sc.edu/NeurodevelopmentalMRIDatabase/ for information and https://www.nitrc.org/projects/neurodevdata/ for access. Details on the online access are provided in Richards, Sanchez, et al. (2015). The data and scripts for all figures presented in the current study are accessible at the Open Science Framework: https://osf.io/pru8y/.

**Funding:** JER: NICHD-R01-HD18942 JER: NICHD-R01-HD18942 JER: NIHCD-R03-HD091464 National Institute of Child Health and Human Development https://www.nichd.nih.gov/ The funder did not play any role in the study design, data collection, analysis, decision to publish, or preparation of the manuscript.

**Competing interests:** The authors have declared that no competing interests exist.

However, there are considerable brain structural changes during infancy through childhood and adulthood [7, 8]. Accurate forward models depend on the use of realistic head models that account for age-related changes in the head [9, 10]. The present study examined age-related changes in DOT sensitivity profiles. We used age-appropriate, realistic head models with extensive coverage of the infancy period (2 weeks through 2 years), two child ages (4, 12 years), and young adults (20 to 24 years).

## The use of Monte Carlo simulations to estimate DOT sensitivity profile

A goal of DOT is to model how hemoglobin changes at the channel space (what is measured) correspond to changes in optical properties within the brain (to be determined). This requires modeling the DOT sensitivity. One approach to solve the forward model is using finite element method (FEM) quantitative algorithms to describe photon propagations through heterogeneous tissue structures such as the human head (NIRFAST; [11]; TOAST++; [12]). Alternatively, Monte Carlo simulations provide numerical solutions to the radiative transport equation that models the photon migration through the head tissues based on fewer assumptions [1, 13]. The Monte Carlo method is computationally demanding but provides more accurate solutions for the forward model [14]. Details of the input parameters and procedures for Monte Carlo simulations are described in [4]. Briefly, the simulations require 1) segmented MRI(s) that define tissue types; 2) optical properties of the tissue types are assigned to each voxel; 3) an injection point and the direction of the photon are specified; 4) an algorithm is applied to describe the path of the photon through the tissues. The steps 3 and 4 are repeated to compute the accumulative photon fluence and the total path length traveled through the head tissues [3]. A computer program is used that takes the input information and simulates the path of injected photons through the head media (e.g., tMCimg; [4]; MCX; [15]; MMC; [16]; MCML; [13, 17]).

DOT sensitivity can be quantified from Monte Carlo simulation outputs. The outputs are fluence distributions of photons travel across the head. The Monte Carlo simulations commonly model fluence distributions for the continuous-wave systems as they are more widely implemented in commercial NIRS instruments than the time-domain and frequency-domain systems [1, 14]. The continuous-wave systems measure fluence at the detector location but do not separately quantify the effects of light absorption and scattering. The DOT sensitivity is conceptualized as the extent to which the detected signal is influenced by changes in optical properties in the given region of tissue [18]. One representation of DOT sensitivity is the spatial sensitivity profile. The profile depicts the fluence distribution of detected photons [4]. It can be quantified by the probability of a photon travelling from a source-optode location to a voxel inside the head ("2-point Green's function"; [3]). A second approach to quantify DOT sensitivity is to use the photon measurement density function (PMDF). The PMDF quantifies optical sensitivity in NIRS source-detector recording by calculating the product of the fluence distribution at the source location and the detector location (PMDF; [9, 19]; or "3-point Green's function; [3]). We will call this the "Source-Detector Channel DOT" (S-D Channel DOT). S-D Channel DOT represents the probability that a photon injected from the source and measured at the detector location has propagated through a given tissue. DOT sensitivity is quantified as each location at which a photon traverses through a given tissue region. "Sampling depth" represents the depth of the photon measured in the head medium.

## DOT sensitivity profile by source-detector separation distances

The DOT sensitivity changes as a function of source-detector separation distances [3, 18, 20, 21]. As the source-detector separation distance increases, the fluence distribution covers wider tissue regions, extends deeper into the brain, and decreases in DOT sensitivity. For example, it

has been found that for 20 mm source-detector separation distance the fluence distribution was largely confined to shallow layers of the head model and about 6% of the total sensitivity was attributed to the brain (gray and white matters) [3, 20]. Sensitivity to the brain increased linearly from 20 mm up to 45 mm separation distance [3, 18, 20, 21]. The increase in sensitivity to the brain was less steep at larger separation distances [3, 20]. There was a trade-off between penetration depth and signal intensity. Signal intensity decreased monotonically and exponentially with increased depth at all separation distances [3]. Wang et al. [21] showed that the signal strength was reduced as source-detector separation distances increased. Hence, the optimal separation distances for measuring brain activity need to be determined based on the consideration of DOT sensitivity, the desired spatial resolution of the measurement regions, and the achievable signal strength for the measurement device.

It is important to use age-specific realistic head models for estimating DOT sensitivity. There is substantial increase in skull thickness [22], CSF volume [23], and global gray matter / white matter growth [7, 8, 23, 24]. There are developmental effects in DOT sensitivity. Age was positively associated with the DOT sensitivity in the frontal and occipital regions in head models of children between 5 to 11 years old [25]. Fukui et al. [10] compared DOT sensitivity by separation distances in neonates (20–24 weeks) and adults. They found that the sensitivity to brain tissues was greater in neonates than adults across all separation distances (10 mm to 60 mm). Additionally, there was an age-by-separation-distance effect. Larger source-detector separation distances (from 40 mm to 50 mm) extended the penetration of the fluence distribution to the gray matter in neonates but did not increase the penetration depth in the adult head model. Age differences in DOT sensitivity may be attributed to anatomical variations across ages. Increased scalp-to-brain distance was related to reduced path length through the brain regions in children [25]. Moreover, greater scalp and skull thickness was associated with reduced sensitivity to the gray matter across source-detector separation distances from 20 mm to 50 mm in adults [20]. These studies concluded that age-related changes in head/brain anatomy and composition affect the DOT sensitivity function. However, these studies are based on a limited range of ages (e.g., neonate; 5 to 11 years) and do not cover the age span across infancy when dramatic development of the head and brain occurs.

## The applications of S-D channel DOT sensitivity profile

The DOT sensitivity profile provides a measure of the scalp-location-to-cortex correspondence. Fu and Richards (under review) estimated the distance between the scalp and the maximum fluence for the S-D Channel DOT fluence. They found that scalp-to-cortex distance increased from infancy to childhood and from childhood to adulthoods. This confirmed that longer source-detector separation distances are required for adults to achieve comparable S-D Channel DOT sensitivity profile.

The S-D Channel fluence distribution can also be used to localize the regions of interest (ROIs) that would be sampled with NIRS. Zimeo Morais et al. [26] used the S-D Channel DOT fluence to compute the specificity of each channel (10–10 electrode pairings) in measuring the corresponding ROIs and non-brain tissues. Fu and Richards (under review) [27] also used the S-D Channel DOT measure to map 10–10 channel locations with underlying ROIs from macro- and micro-structural atlases. Such channel-to-ROI look-up tables facilitate the optimization of channel array design to maximize DOT sensitivity to user-specified ROIs [26, 28] and help to localize the possible ROIs that generated the fNIRS signals [29–31]. Existing tools for designing channel placements based on ROIs estimates DOT sensitivity using adult head models. Hence, it is important to evaluate whether the estimations from adult head models can be applied to developmental samples.

### The present study

The current study examined age-related variations in DOT sensitivity profiles. Our primary contribution was to use existing methods for DOT sensitivity across the period of infancy (2 weeks through 24 months) with narrow age bands, and compare these to older children (4, 12 years) and adults (20–24 years). We used the MCX photon migration simulation program to model photon propagation through the segmented tissues in the individual head MRIs (Fang & Boas [15]; see Supporting Information for further information). We examined S-D Channel DOT fluence as a function of sampling depth. We also depicted how the DOT sensitivity profile changes across source-detector separation distances. We expect to replicate existing findings in adults. That is, the fluence strength will decrease with increased depth at all source-detector separation distances. The shape of the DOT sensitivity profile will change from short to longer separation distances. The study will provide novel evidence revealing developmental changes in how the DOT sensitivity profiles vary across separation distances. The present study has important implications in the design of age-specific channel placements for NIRS recordings and make anatomical interpretations of stand-alone NIR data across a wide range of age groups.

## Materials and methods

### Participants

The participants were 1058 typically developing participants ranging from 2 weeks to 24 years of age. The MRIs were obtained from open-access databases and a local scanning facility. Details of the sample are described in [27]. Table 1 presents the number of MRIs for the open access databases, separately for age and gender. The sample ages were narrowest in the infancy period (1-, 1.5-, 3-, or 6-month intervals from 2 weeks through 2 years) and included exemplar

**Table 1. Demographical information of study participants by age group, sex, and data source.**

| Participant Information | | | Data Source | | | | | |
|---|---|---|---|---|---|---|---|---|
| Age Group | Total *N* | Female *N* | ABIDE *N* | BCP *N* | EBDS *N* | IBIS *N* | MCBI and Collaboration Sites *N* | PING *N* |
| 2 Weeks | 41 | 24 | 0 | 3 | 38 | 0 | 0 | 0 |
| 1 Month | 96 | 40 | 0 | 17 | 79 | 0 | 0 | 0 |
| 2 Months | 68 | 40 | 0 | 8 | 60 | 0 | 0 | 0 |
| 3 Months | 38 | 21 | 0 | 24 | 0 | 0 | 14 | 0 |
| 4.5 Months | 54 | 29 | 0 | 41 | 0 | 0 | 13 | 0 |
| 6 Months | 74 | 35 | 0 | 0 | 0 | 60 | 14 | 0 |
| 7.5 Months | 61 | 17 | 0 | 0 | 0 | 49 | 12 | 0 |
| 9 Months | 60 | 35 | 0 | 48 | 0 | 3 | 9 | 0 |
| 10.5 Months | 42 | 21 | 0 | 40 | 0 | 0 | 2 | 0 |
| 12 Months | 109 | 36 | 0 | 0 | 0 | 89 | 20 | 0 |
| 15 Months | 78 | 41 | 0 | 63 | 0 | 8 | 7 | 0 |
| 18 Months | 76 | 31 | 0 | 36 | 0 | 8 | 32 | 0 |
| 2 Years | 66 | 22 | 0 | 0 | 0 | 65 | 1 | 0 |
| 4 Years | 24 | 9 | 0 | 0 | 0 | 0 | 10 | 14 |
| 12 Years | 37 | 14 | 9 | 0 | 0 | 0 | 28 | 0 |
| 20–24 Years | 134 | 77 | 0 | 0 | 0 | 0 | 134 | 0 |

*Note*. ABIDE = Autism Brain Imaging Data Exchange; BCP = Baby Connectome Project; EBDS = Early Brain Development Study; IBIS = Infant Brain Imaging Study; MCBI = McCausland Center for Brain Imaging; PING = Pediatric Imaging, Neurocognition, and Genetics Data Repository

ages in children and adolescent ages (4, 12 years) and adult (20–24 years). All studies had institutional review board approval and informed consent. All data were collected specifically for research purposes. Data were completely anonymized and deidentified for processing and analysis. The University of South Carolina Institutional Review Board approved the current study (including data collection at the MacCausland Center for Brain Imaging (MCBI) and the use of data from all open-access databases).

## MRI sequences

The present study utilized T1-weighted (T1W) and T2-weighted (T2W) scans from each collection site. Details of the MRI acquisition protocols have been described in literatures on the Neurodevelopmental MRI Database [32–37]. All MRIs were converted to NIFTI compressed format with 32-bit floating point resolution. Bias-field inhomogeneity correction (N4 algorithm) was performed on the extracted T1-weighted images [38, 39].

## MRI preprocessing and segmentation

First, the brains were extracted from the whole-head MRI volume in a procedure adapted from the FSL VBM pipeline [40]. The T1W volume for each participant was registered to an age-appropriate average MRI template. The MRI templates came from the Neurodevelopmental MRI database [35–37]. The brain from the average template was transformed into the participant MRI space and used a mask on the head volume. The extracted masked data was then used with the FSL brain extraction tool program [41, 42]. Each brain was visually inspected and manually modified if necessary. Second, each head MRI volume was segmented into 9 or 10 media types: gray matter (GM), white matter (WM), cerebrospinal fluid (CSF), non-myelinated axons (NMA), other brain matter, skin, skull, air, eyes, and other inside skull material. Details of the segmentation methods are presented in the Supporting Information. The segmented regions were assembled into a single MRI volume we will refer to as the "segmented head MRI volume". Fig 1A shows a 3D rendering of the T1W volume from a 3-month-old infant with a cutout revealing the segmented MRI volume. The realistic head model represents the geometry of the head and allow us to differentiate optical properties of different tissue types.

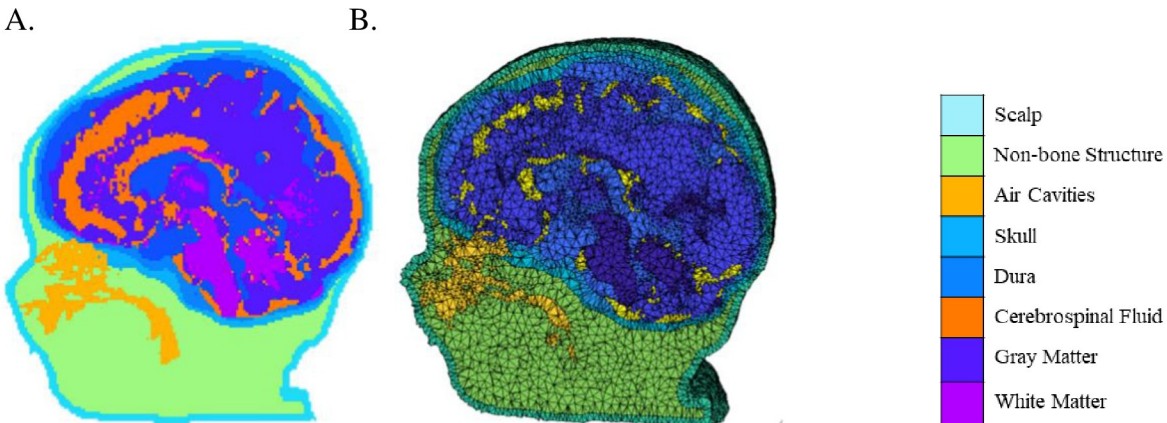

**Fig 1. Segmented head MRI volumes.** The examples were taken from the same 3-month-old infant MRI. A. The segmented head model. Aqua is the scalp, green is the non-bone structure (muscle, sinew, fat), gold is the nasal and mouth air cavities, turquoise is the skull, light blue is the dura, orange is the cerebrospinal fluid, dark blue is the gray matter, and purple is the white matter. B. The segmented head model with dense finite element (FE) mesh.

## Mesh generation

A "finite element" (FE) tetrahedral mesh was constructed from the segmented head MRI volume. Fig 1B displayed the dense meshes. They were produced using the iso2mesh toolbox with CGAL 3.6 mesh program ("v2m"; [43]). The FE volumetric meshes have nodes that represent the voxel locations for the tetrahedra, a 4-element matrix representing the corners of each tetrahedron, and a vector representing the media type from the segmented head MRI volume. A mesh was generated for each segmented head MRI volume. The number of nodes, elements, and tetra volume were calculated for the infants (2 weeks through 2 years), children (4 and 12 years), and adults (Fig 1 in S1 File). We will refer to this as the "segmented FE mesh". The dense "segmented FE mesh" was used for MCX [15] to find a segment element that is closest to an optode position.

## Scalp locations

**Virtual optodes placement.** The locations for the 10–10 and 10–5 electrode systems [44] were constructed on each head MRI volume. These standard locations were used for virtual optodes placement. First, we manually marked cranial fiducial points using MRIcron [45, 46]: nasion (Nz), vertex (Vz), inion (Iz), left preauricular point (LPA), right preauricular point (RPA), left mastoid (LMa), and right mastoid (RMa) [47]. The coordinates of the fiducials were transferred on to the scalp mesh. Next, we calculated 81 virtual positions based on the "unambiguously illustrated 10–10 system [44]. Details for constructing the 10–10 locations are described in [47] and the Supporting Information. We simulated a total of 358 optodes on 10–5 locations by calculating center points between 10–10 positions. The optode positions were also computed for the average MRI templates. Fig 2 illustrates the virtual optodes placements at 10–10 and 10–5 electrode positions.

**Source-detector channels, all optode pairs.** Source-detector channels were defined in two ways for the analyses. *Source-Detector Channels*: Source-detector channel locations were defined using optode combinations centered on each 10–10 electrode location. The 10–10 electrode locations were centered between surrounding adjacent pairs of 10–10 or 10–5 electrode locations. There were 251 source-detector channels formed with adjacent 10–10 electrodes, $Mean_{Separation}$ = 58.0 mm; $SD$ = 14.2, and 251 channels formed with adjacent 10–5 electrodes, $Mean_{Separation}$ = 28.9 mm; $SD$ = 7.1. The channel locations were used to estimate "S-D Channel DOT fluence" described below. *All Optode Pairs*: Some channel locations were defined from the combination of all 10–5 electrode positions for the source-detector separation distance analyses. The path from each optode to all other optodes was traced on the scalp and the distance of the path recorded (63,903 paths). The channel location was defined at the half-way point in this path. The optode pairs, scalp path distances, and channel locations were recorded. Source-detector separation distances were calculated as half of the scalp path distances. The separation distances were used for the S-D Channel DOT sensitivity profile analysis described below.

## Photon migration simulations

The DOT fluence values were estimated with photon migration simulations. We modeled photon migration through brain tissues using the Monte Carlo eXtreme package (MCX; [15, 48]). The package implements an GPU-accelerated, voxel-based Monte Carlo simulation algorithm. The inputs of Monte Carlo simulations include a segmented head model that defines the tissue types. Five tissue types are typically identified: the scalp, skull, cerebrospinal fluid (CSF), GM, and WM [3, 10, 18]. The voxel-based MCX simulations do not require "segmented FE mesh" [15]. The default voxel size $1 \times 1 \times 1 mm^3$ was used. We launched $10^8$ photons for the time

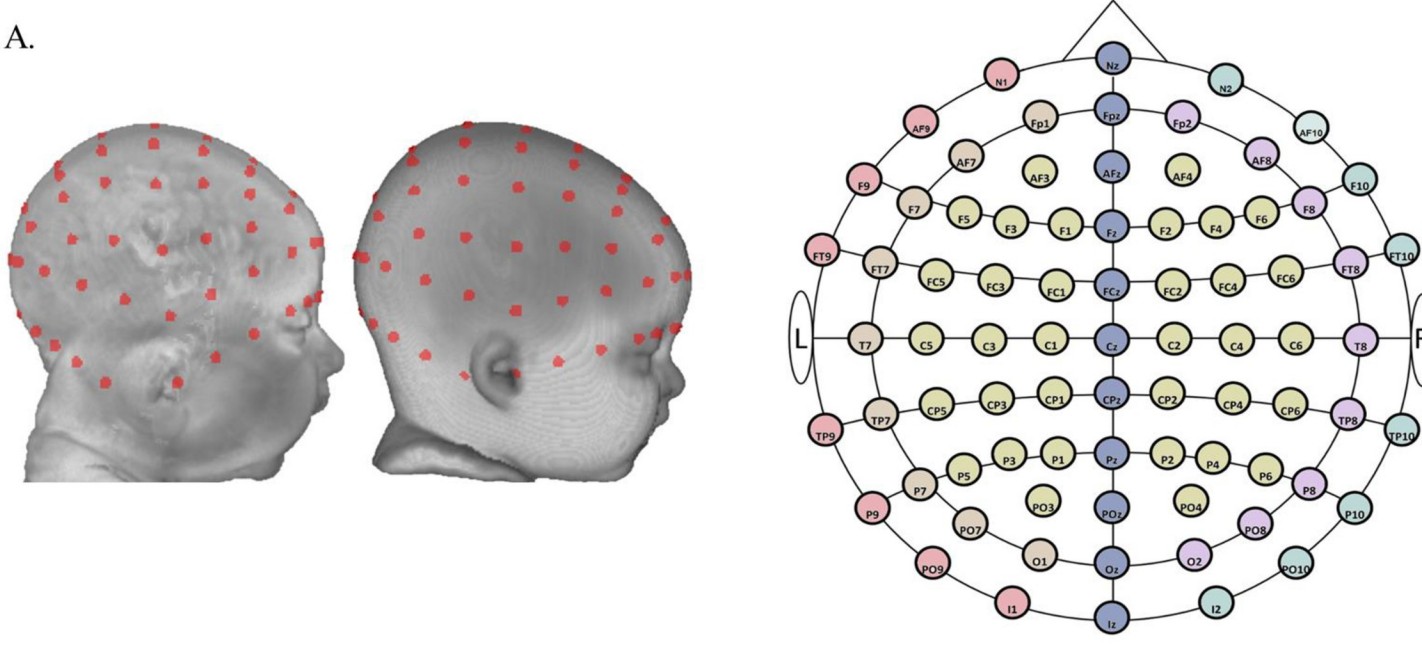

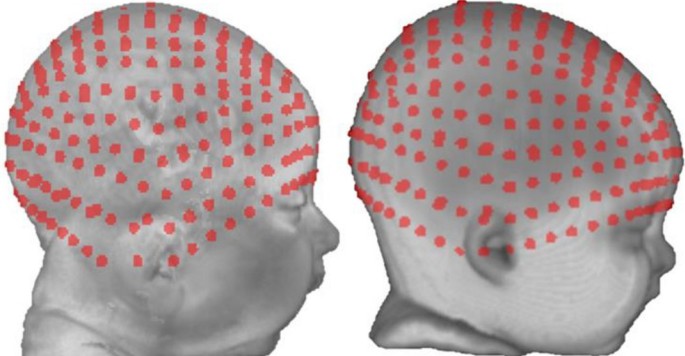

**Fig 2. Virtual optodes placement.** A. Virtual optodes placement on the standard 10–10 electrode system. From left to right: a three-month individual head model, an average template for three-month-old's, and a two-dimensional layout of the 10–10 system). Ten-ten electrodes were divided into six groups for visualization purposes. The six groups were color-coded on the two-dimensional schematic of the 10–10 system. Group 1: electrodes on the central curve (Nz-Cz-Iz). Group 2: electrodes on the left curve between Nz and Iz (N1-LPA/T9-I1). Group 3: electrodes on the right curve between Nz and Iz (N2-RPA/T10-I2). Group 4: electrodes on the left curve between Fpz and Oz (Fp1-T7-O1). Group 5: electrodes on the right curve between Fpz and Oz (Fp2-T8-O2). Group 6: the remaining electrodes enclosed by the central curve, the left and right curves between Fpz and Oz. B. Virtual optodes placement on the 10–5 electrode system. From left to right: the same three-month individual head model and the three-month average template.

window of 0 to 5 nanoseconds. The fluence resolution was 50 time-gates [26]. The wavelength was set at 690 nanometers. Photons were sent from the 358 10–5 positions. The optical properties of the 10 tissue types that are specified include the absorption coefficient, the scattering coefficient, the anisotropy coefficient, and the index of refraction.

There is no consensus on the optical properties in the literature [26]. The values differ considerably across studies [2–4, 19, 25, 26, 49]. Different values may be needed for adult and infant-child participants (cf. [10, 19, 25]). S1 Table summarizes optical properties used in several adult and infant-child studies. The table shows that studies commonly set optical

**Table 2. Optical properties used in Monte Carlo photon migration simulations.**

| Tissue Type | $\mu_a$ (mm$^{-1}$) | $\mu_s$ (mm$^{-1}$) | g | N | Sources and Rationales |
|---|---|---|---|---|---|
| White matter | 0.07 | 40.1 | 0.85 | 1.37 | [50, 51] |
| Gray matter | 0.02 | 8.4 | 0.90 | 1.37 | [50, 51] |
| CSF | 0.0004 | 1 | 0.99 | 1.37 | [16, 49, 50, 52]; mimic the optical properties of water. |
| Dura | 0.0101 | 80 | 0.99 | 1.37 | [50] for skin and non-brain tissues |
| Skull | 0.0101 | 100 | 0.99 | 1.37 | [2, 49, 50] |
| Skin | 0.0101 | 80 | 0.99 | 1.37 | [50] for skin and non-brain tissues |
| Muscle | 0.0101 | 80 | 0.99 | 1.37 | [50] for skin and non-brain tissues |
| Eyes | 0.0004 | 1 | 0.99 | 1.37 | Mimic the optical properties of water. |
| Nasal Cavity | 0.0101 | 80 | 0.99 | 1.37 | [50] for skin and non-brain tissues |
| Non-myelinated Axons | 0.07 | 40.1 | 0.85 | 1.37 | Mimic the optical properties of WM for infants at 12 months of age or younger. |

$\mu_a$ is the optical absorption coefficient; $\mu_s$ is the scattering coefficient, $g$ is the anisotropy coefficient, and $N$ is the index of refraction.

The wavelength was set at 690 nanometers.

properties for scalp and skull, CSF, GM, and WM. Our input values for the optical properties of the head media and the sources of these values were shown in Table 2. They were based on values used in [50], which primarily came from [16, 49, 51].

The Monte Carlo simulations estimate fluence distribution [3, 4]. The photons were injected at the optode position. The "flux" strength of the photons, or "fluence" of the sum photon flux, decreased monotonically due to absorption and scattering as photons traveled through brain tissues. The simulations accounted for absorption, scattering and reflection based on the specified optical properties [4, 15]. Each simulation continued until the photon exited the media, was absorbed by the media, or after 5 nanoseconds. The output of the simulations included the photon fluence at each nanosecond for each voxel from the injected photons. The number ranged from zero to the total number of injected photons. We summed this value across the 5 nanoseconds of the simulation. This value represents the fluence of the simulation. We took the log (fluence) as our number representing the DOT sensitivity for each voxel.

## DOT sensitivity analyses

The output from the Monte Carlo simulation contained the fluence across the entire MRI volume separately for each optode. We simulated the path from a source-optode to a detector-optode. This was done by multiplying the source-optode fluence distribution by the detector-optode fluence distribution. We will refer to this as "S-D Channel DOT fluence". It is a unitless measure that represents the sensitivity of the DOT measure for detecting changes in optical properties at a given point inside the head medium (PMDF; [9, 19]; or "3-point Green's function; [3]). An ad-hoc method has been used in some studies to divide the product by the estimated value of the photons from the source in the vicinity of the detector [50]. This step was not taken in the current study. Fig 3 shows the S-D Channel DOT fluence plotted on an MRI for a single source-detector channel. This figure shows a similar monotonic decrease in fluence strength as photons traveled deeper into the head tissues.

## S-D channel DOT sensitivity profile by source-detector separation distances

We examined how the of S-D Channel DOT sensitivity profile changed as a function of source-detector separation distance. The S-D Channel DOT fluence was extracted voxel by

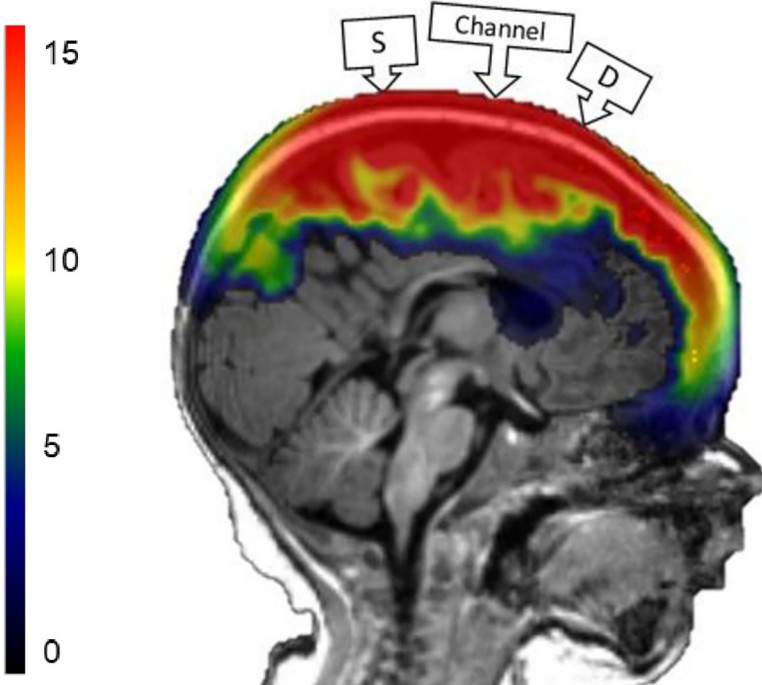

**Fig 3. Source-detector (S-D) channel DOT fluence distribution.** Monte Carlo photon migration simulations were used to estimate fluence distributions. The red area represents greater fluence.

voxel. We recorded the fluence value and the distance (depth) from the channel location to the voxel with the fluence estimation. We refer the distance from the channel to the location of the fluence value inside the head model as "sampling depth" for short. The S-D Channel DOT sensitivity profile was defined as the fluence value at the voxel as a function of the sampling depth. The target source-detector separation distances for testing were from 10 mm to 60 mm by 5 mm increments. They were selected from "all optode pairs". S2 Table lists the minimum and maximum separation distances for each target separation distance for all age group. Fig 2 in S1 File displays the mean number the source-detector channels with the target separation distances by age group. For each participant and each channel with the target separation distance (e.g. 400 channels with 10mm separation distance), we computed the S-D Channel DOT fluence value at each sampling depth. The fluence values were then averaged across channels by sampling depth (1 to 100), target separation distance, and participant. We additionally assessed the variance of the sensitivity profile by examining the standard error of the mean fluence value as a function of sampling depth. The shape of the S-D Channel DOT fluence distribution was quantified by the half-width half-maximum (HWHM; [53]). The HWHM identified the sampling depth at which the fluence first dropped below half of the maximum value. We referred to this as the HWHM location.

## Results

### S-D channel DOT sensitivity profiles by source-detector separation distances

We examined the S-D Channel DOT fluence as a function of the depth between channel location and the voxel sampled in the fluence distribution ("sampling depth"). This was done by

selecting 10–5 location optode pairs with separation distances from 10 mm to 60 mm by 5 mm increments and calculating the S-D Channel DOT fluence for each pair and averaging over pairs of the same separation distance. Fig 4 displays examples of the S-D Channel DOT sensitivity profiles plotted by channels (separate lines) and separation distances (panels) for one individual from four different ages at 20 mm and 50 mm separation distances (more examples are displayed in Fig 3A and 3B in S1 File). The sensitivity profiles across the four individuals showed greater peaks and steeper slopes for channels at 20 mm than 50 mm separation distance. The between-channel variance showed a systematic decline from the infants to the adults. The variance increased from 20 mm to 50 mm separation distances for the infant groups whereas it remained stable for the adult group.

The sensitivity profiles changed with separation distances across age groups. Fig 5A shows the S-D Channel DOT fluence as a function of sampling depth separately for the target source-detector separation distances, averaged across channel locations, and separately for four age bins. We grouped the samples to four age bins for averages based on preliminary data visualization: 2 weeks through 4.5 months; 6 months through 12 months; 15 months through 2 years; 4, 12, and 20–24 years. The sensitivity profile changed systematically with the source-detector separation distance. The peak of the sensitivity function was largest at the smallest separation distance and decreased as source-detector separation distance increased. However, the fluence value dissipated more quickly at the shorter source-detector separation distances than the longer separation distances, with the distributions becoming flatter across increasing source-detector separation distances. Compared to the sensitivity profile at shorter separation distances, the same fluence value was carried to greater sampling depth at larger separation distances.

The patterns of age-related differences in the sensitivity profiles were relatively consistent across separation distances. Fig 5B–5D, and S1 File in Fig 4 show the S-D Channel DOT fluence sensitivity functions for the 20 mm, 30 mm, and 50 mm source-detector separation distances, respectively, and separately for the testing ages. There were consistent age-related changes in the shape of the sensitivity function over the source-detector separation distances. These are most easily seen in the profile for the 50 mm separation distance (Fig 5D and S1 File in Fig 4C). The line graph in the Main Text shows that the 2-week-to-4.5-month age groups (Fig 5A solid line, aqua color) had a relatively high peak and rapid decline, the 6-month-to-12-month (Fig 5A solid line blue color) and the 15-month-to-2-year age groups (Fig 5A dashed line red color) had a smaller peak value and less rapid decline, i.e., flatter shape. The bar graph in the Supporting Information additionally shows that the sensitivity function at the 50 mm separation distance had a more flattened shape across all age groups. Age-related differences were greater at between 0 to 40 mm sampling depth. Together, the figures indicate that the age differences were similar across all source-detector separation distances but became greater with increasing source-detector separation distances.

**Summary.** Our findings showed that the DOT sensitivity profiles display similar characteristics across age groups but also important age-related differences. The S-D Channel DOT fluence decreased exponentially in all age groups as the light traveled deeper into the head tissues. The sensitivity profiles for the 20-to-24-year-olds showed that as the separation distances increased, the peak of the fluence value decreased, but the fluence strength declined less rapidly as the light traveled through the tissues. These patterns confirmed the DOT sensitivity profiles found in previous studies using adult head models [3, 18, 20, 21]. Older infants (6 months to 2 years) displayed different DOT sensitivity profiles than young infants (2 weeks to 4.5 months) and adults, and the age-group differences are more discernible in larger source-detector separation distances.

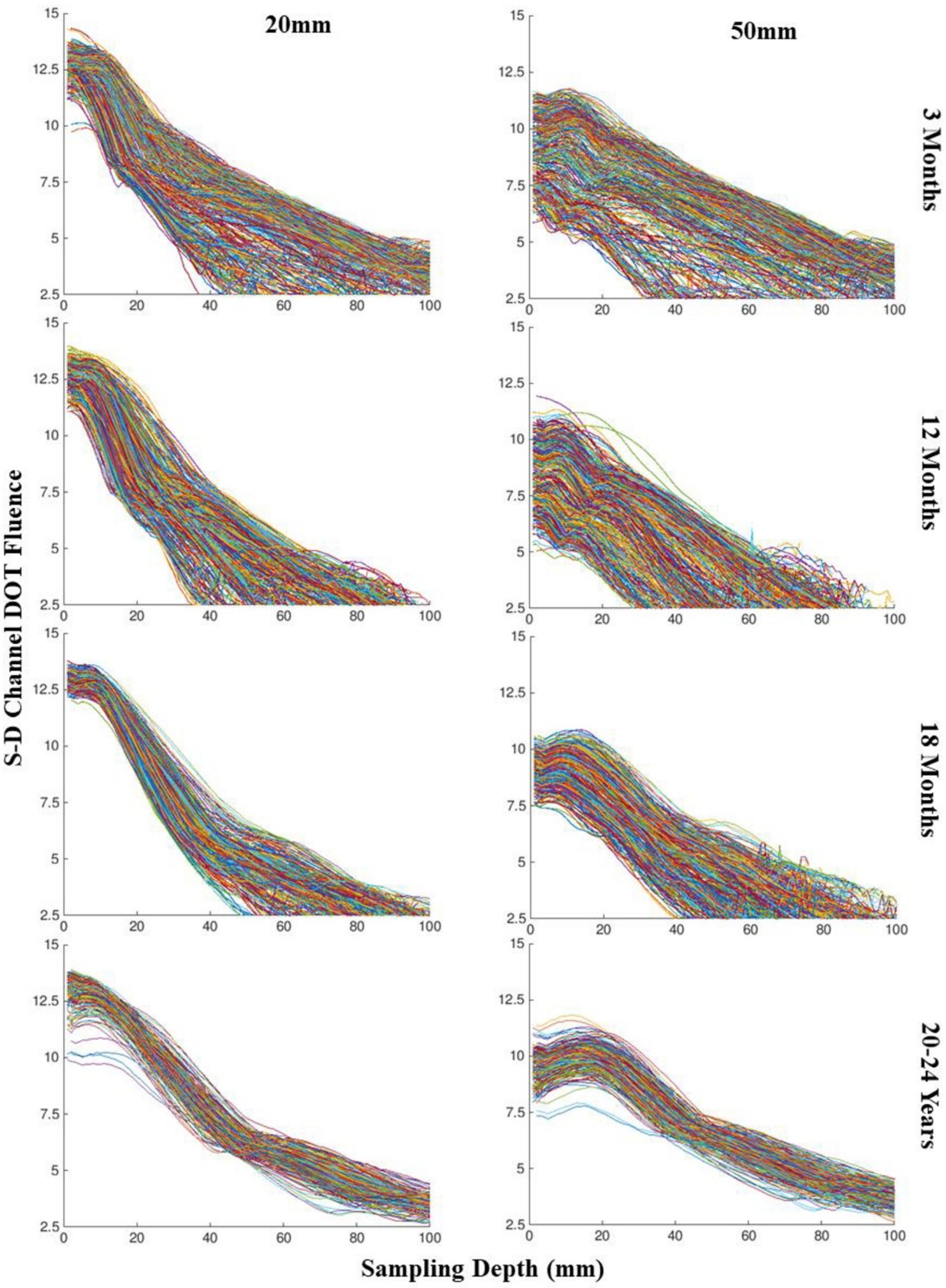

**Fig 4. Source-detector (S-D) channel DOT fluence sensitivity profile by channels.** Four example individuals were selected, one from each age category. The S-D Channel DOT fluence value was plotted as a function of the sampling depth separately for channels at the source-detector separation distances for 20 mm and 50 mm for everyone. For each participant and each channel with the target separation distance, we computed the S-D Channel DOT fluence and the distance from the channel location to the voxel with the fluence estimation ("sampling depth").

## The variance of S-D channel DOT sensitivity profiles

We also examined the between-channel variance of the S-D Channel DOT fluence as a function of sampling depth. Fig 4 shows the average between-channel variance of the sensitivity profiles across separation distances, which apparently differed across ages and showed a larger change with source-detector separation distance for infants than adults. We further examined the variance by computing the average standard error of the mean S-D Channel DOT fluence value. Fig 6A displays the average of the standard error as a function of sampling depth for

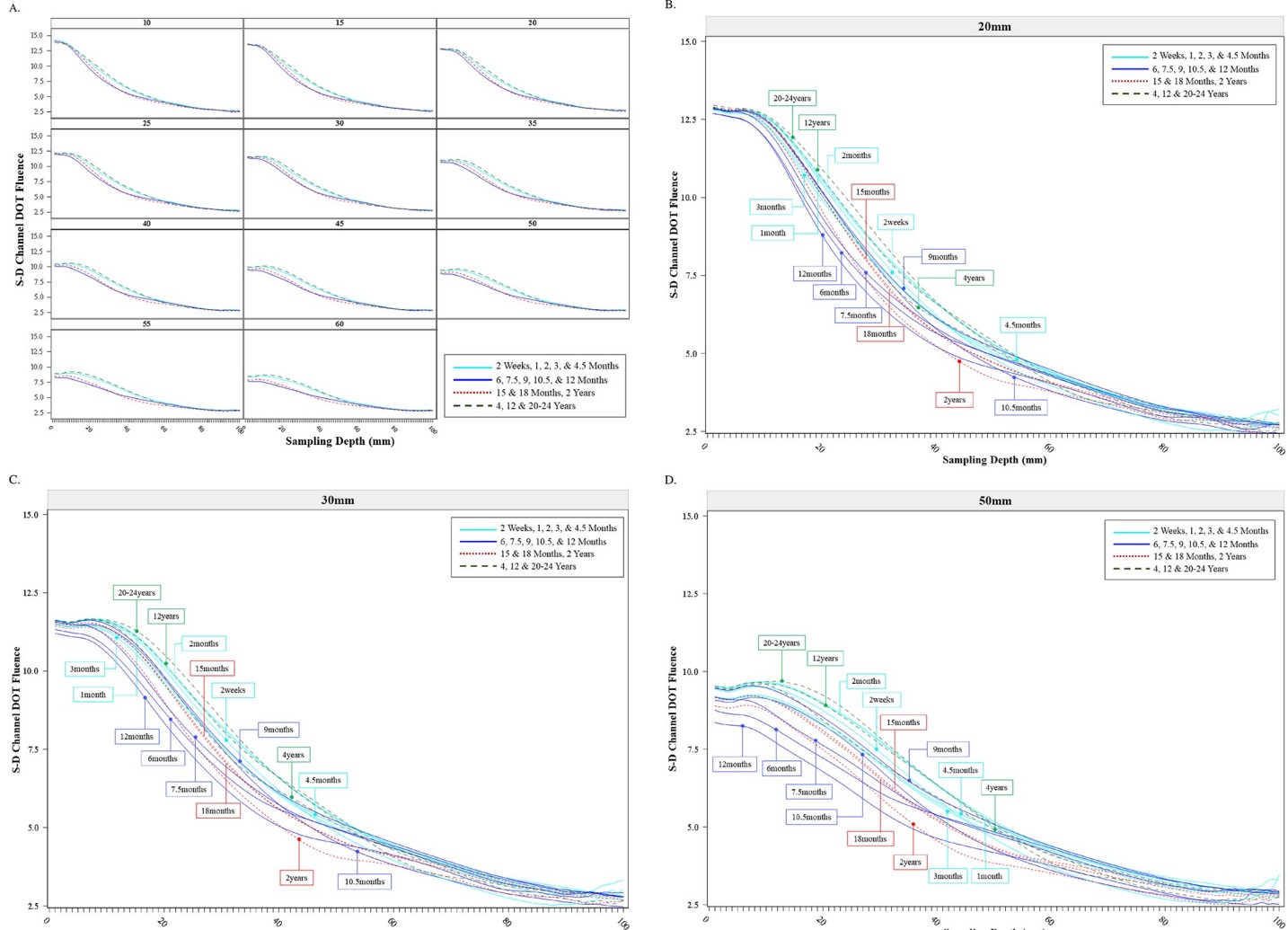

**Fig 5. Source-detector (S-D) channel DOT fluence sensitivity profile by age groups.** The S-D Channel DOT fluence value was plotted as a function of the sampling depth, defined as the distance (depth) from the channel location to the voxel location in the head model where the S-D channel DOT fluence was measured. A. S-D Channel DOT fluence sensitivity function at each target source-detector separation distances. For visualization, we created four age categories: 2 weeks to 4.5 months, 6 months to 12 months, 15 months to 2 years, and 4 years to 20–24 years. Mean values by each age category are displayed. B. S-D Channel DOT fluence sensitivity function by individual age categories at 20 mm source-detector separation distance. C. S-D Channel DOT fluence sensitivity function by individual age categories at 30 mm separation distance. D. S-D Channel DOT fluence sensitivity function by individual age categories at 50 mm separation distance.

A.

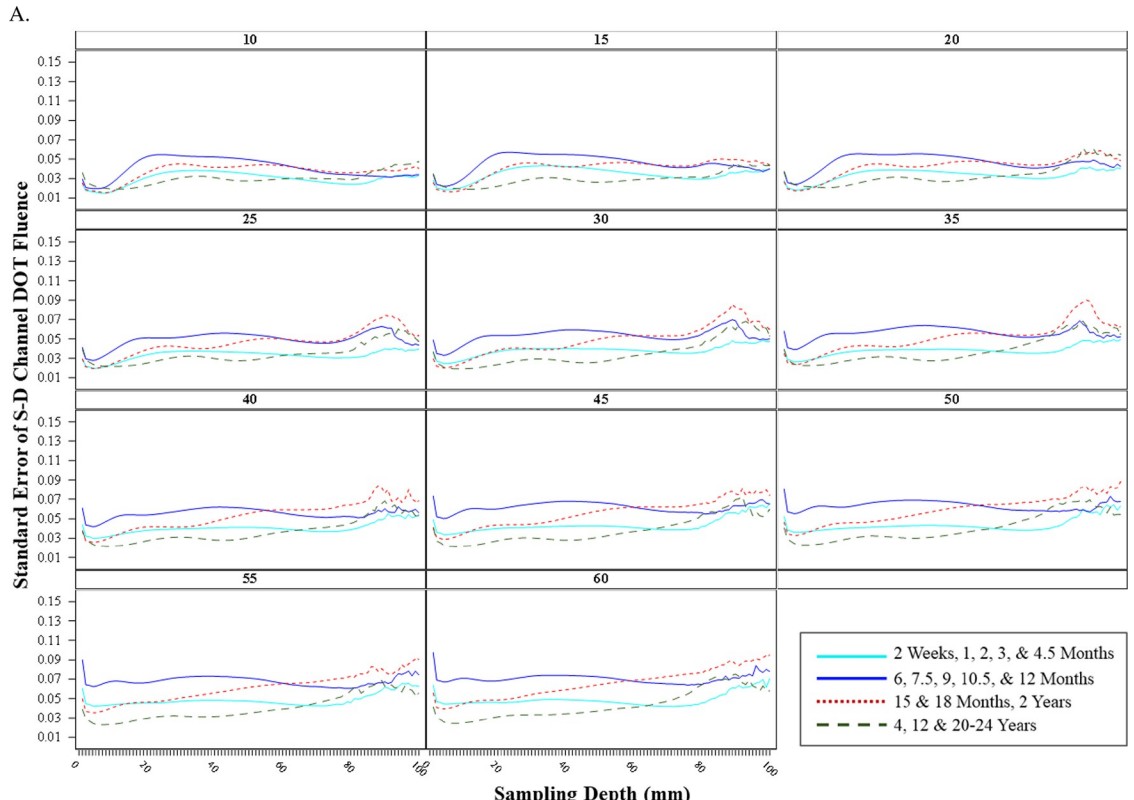

B.

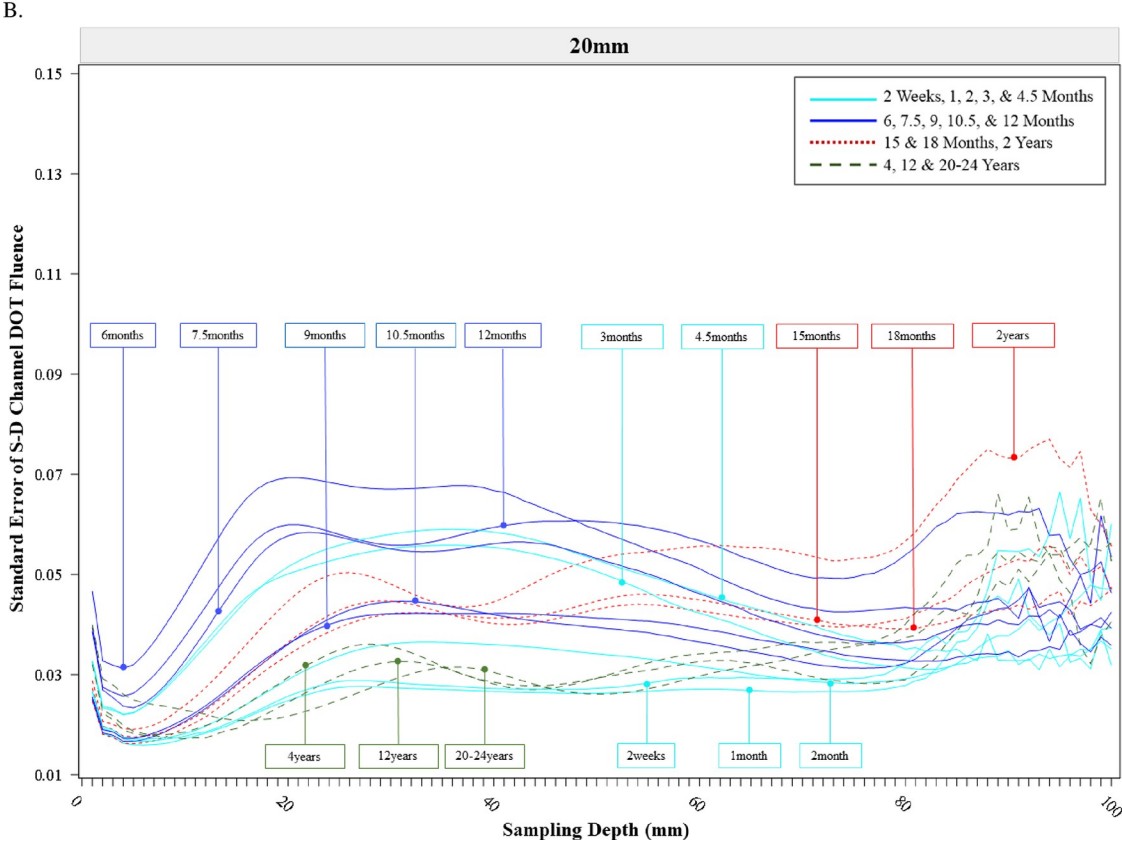

**Fig 6. Variance of source-detector (S-D) channel DOT fluence sensitivity profile.** This is the standard error of the S-D Channel DOT fluence value as a function of the sampling depth, defined as the distance (depth) from the channel location to the voxel location in the head model where the S-D channel DOT fluence was measured. A. Standard error of the S-D Channel DOT fluence sensitivity function averaged by the age categories. B. Standard error of the S-D Channel DOT fluence sensitivity function by individual age groups at 20 mm source-detector separation distance.

each separation distance. The average variance as a function of sampling depth increased at separation distances greater than 35 mm for the younger infants (2 weeks to 12 months). The average variance became larger with increased sampling depth for the infants between 15 months to 2 years of age at separation distances greater than 20 mm. The average variance across sampling depth significantly increased at 55 mm and 60 mm separation distances for the 15-month-to-2-year group. The average variance across sampling depth remained relatively stable for the child and adult group. Fig 6B and S1 File in Fig 5 show the standard error of the mean S-D Channel DOT fluence value as a function of sampling depth separately by the testing age groups at 20 mm separation distance. The variance across sampling depth was the greatest for infants aged between 3 months to 2 years relative to the children and adults. The youngest infants (2 weeks, 1 month, 2 months) showed relatively small variance across separation distances. We present similar figures for 30 mm and 50 mm separation distances in Fig 6A and 6B in S1 File. These figures show that the variance as a function of sampling depth was greater at larger separation distances for all age groups. The age-related differences in variance were enhanced at 50 mm separation distances. The increase in variance at larger separation distance is more discernible for infants between 3 months to 2 years. The patterns of age-group differences in variance of the sensitivity profile across separation distances were consistent with the individual-subject plots displayed in Fig 4 and S1 File in Fig 3.

**Summary.** Between-channel variance was relatively stable for the children and adults. The variance increased as a function of source-detector separation distance in infant groups. The increase was most observable in older infants (6 months to 2 years).

## The shape of S-D channel DOT sensitivity profiles

The last analysis examined the half-width half-maximum (HWHM) location for each S-D Channel DOT fluence distribution. Fig 7 shows the HWHM location as a function of the source-detector separation distance separately for the testing ages (Fig 7A) and for four separate age groupings (Fig 7B). Fig 7 in S1 File displays the HWMH location as a function of age groups, separately for separation distances. The HWHM increased as a function of the source-detector separation distance for all age groups. This increase quantifies the apparent change in steepness in the S-D Channel DOT fluence sensitivity functions over source-detector separation distances (Fig 5A). The change in HWHM across source-detector separation distance implies that there was a gain in sensitivity to deeper cortex over the source-detector separation distance, though the fluence strength systematically decreased with increasing source-detector separation distance (e.g., Fig 5A). The patterns of the HWHM location over age parallel the age differences found in the sensitivity profile (e.g., Fig 5A). The mean HWHM locations across all separation distances were greater in the youngest infants (2-week-to-4.5-month group, Fig 7A and 7B solid line, aqua color), children and adults (Fig 7A and 7B green dashed line), and at smaller levels in the 6-month-to-12-month group (Fig 7A and 7B solid line blue color) and the 15-month-to-2-year group (Fig 7A and 7B dashed line red color). These differences were consistent with the age differences found in the sensitivity profile (e.g., Fig 5A). The larger HWHM location values imply the DOT sensitivity to deeper cortical areas for the youngest infants, children, and adults.

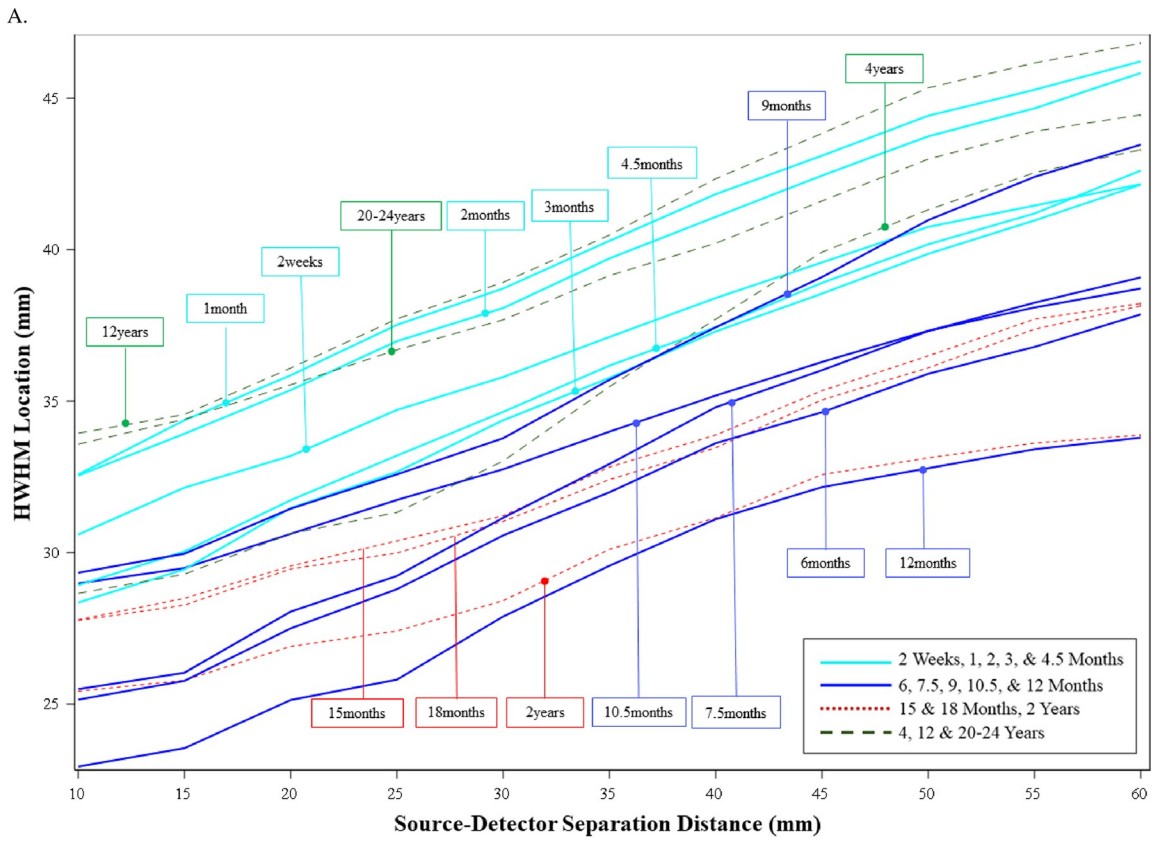

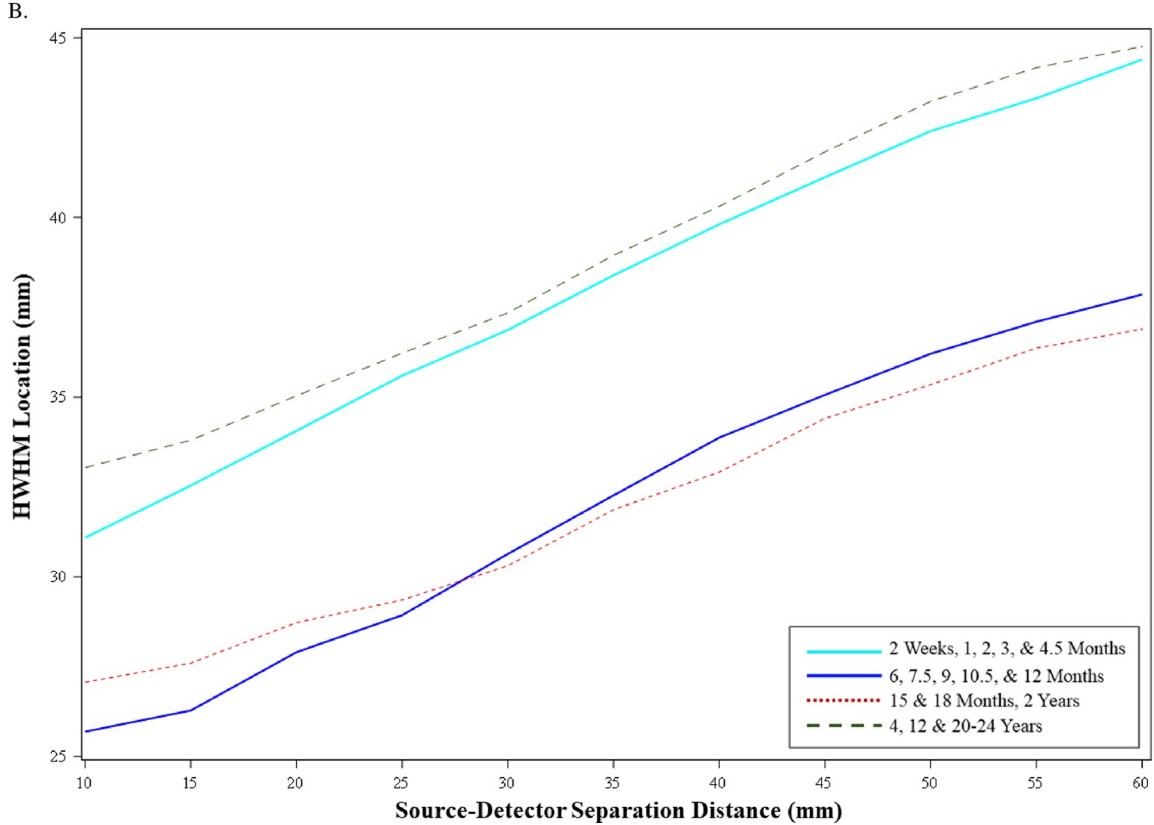

**Fig 7. Half-width half-maximum (HWHM) locations of source-detector (S-D) channel fluence as a function of source-detector separation distances.** The figure shows age-related differences in how the shape of the S-D Channel DOT sensitivity profile changes with separation distances. The HWHM location was defined as the point in the fluence distribution at which the fluence first dropped below half of the maximum value. A. HWHM locations by source-detector separation distances for individual age groups. B. Mean HWHM locations by source-detector separation distances for the four age categories: 2 weeks to 4.5 months, 6 months to 12 months, 15 months to 2 years, and 4 years to 24 years.

**Summary.** The shape of DOT sensitivity profiles corroborated existing evidence that the fluence distribution extends deeper into the head tissues as the source-detector distance increases [3, 18, 20, 21]. This pattern was found in all age groups. The important age-related difference was that the same proportion of fluence strength carried deeper into the head tissues for the young infant, child, and adult groups across all separation distances.

## Discussion

The present study examined age differences in DOT sensitivity profiles. We extended existing photon migration simulation methods to realistic head models from infants ranging from 2 weeks to 2 years, children (4, 12 years) and adults. The DOT sensitivity profiles had a common shape across source-detector separation and ages. There were expected differences in the S-D Channel DOT sensitivity profile across source-detector separation distances, with larger separation distances being more sensitivity to greater cortical depths but showing decreasing signal strength and larger variance in the fluence measure. A noticeable age-group difference was that the fluence distribution penetrated greater depth with the same level of signal loss in the younger infant (2 weeks to 4.5months) than the rest of the infant age groups. The shape of S-D Channel DOT sensitivity profile was similar between the younger infant groups and child and adult groups.

### S-D channel DOT sensitivity profile by source-detector separation distances

The present study showed that the S-D Channel DOT sensitivity profiles changed systematically as a function of source-detector separation distance for all age groups. The study enhanced the DOT sensitivity estimation accuracy by using realistic FEM head models with refined segmentation of tissue types. The S-D Channel DOT sensitivity profiles depict S-D Channel DOT fluence as a function of sampling depth in the fluence distribution. The peak value of the S-D Channel DOT fluence was larger at shorter separation distances but showed rapid decay as sampling depth increased. Alternatively, at larger separation distances there was a smaller peak value but a less rapid decline with increased sampling depth. The shape of the sensitivity profile was characterized by measuring the point in the fluence distribution where the fluence value first dissipated below half of the maximum value (i.e., "HWHM location"). The HWHM location increased with source-detector separation distance for all age groups. This indicates that there is a gain in sensitivity to deeper cortical structures with increased separations at the expense of decreased signal strength. We replicated the robust characteristics of the DOT sensitivity profiles found in existing studies that modeled photon migrations in head models from infants [10] and adults [3, 18, 20, 21]. The trade-off between the gain in sensitivity to deeper cortical regions and loss in signal strength and spatial resolution needs to be considered in the design of source-detector separation distances.

The sensitivity profiles showed relatively consistent age-related differences across source-detector separation distances. Young infant groups (2 weeks to 4.5 months) displayed a sensitivity profile that was significantly different from the older infant groups (6 months to 2 years). The sensitivity profiles for the young infant groups at all separation distances were

characterized by higher peaks and steeper decreases of S-D Channel DOT fluence values with increased sampling depth. Their sensitivity profiles displayed relatively stable variance across separation distances. They had greater HWHM locations than older infant group at all separation distances. This means that the same fluence level extended deeper into the cortex in the younger infant groups. In contrast, the sensitivity profiles for the older infant groups were characterized by smaller peaks and more gradual declines with increased sampling depth. Their sensitivity profiles showed greater variance at larger separation distances. The slopes of increase in HWHM location across source-detector separation distance were similar across age groups. This suggests that the gain in S-D Channel DOT sensitivity by increasing separation distances is comparable across age groups.

Developmental changes in the morphology of the head and the brain may contribute to the age-related differences in S-D Channel DOT sensitivity profiles. Infants have thinner extracerebral layers (scalp and skull) and thicker CSF than adults [19]. The skull of younger infants has greater curvature and is more flexible. Younger infants might have a thicker CSF layer in the back of their heads due to the head positioning during the MRI scan [54]. The low scattering CSF may confine the S-D Channel DOT fluence distribution to shallower layers of the cortex especially at shorter source-detector separation distances. Fu and Richards (under review) [27] showed that the distance between the scalp channel location and the maximum fluence in the S-D Channel DOT fluence distribution was relatively shorter for the 2-week to 2-month groups. A direction for future research is to characterize morphological changes in specific tissue layers that are linked to age-related differences in S-D Channel DOT sensitivity profiles across separation distances.

## Applications of age-specific DOT sensitivity profiles

The age differences highlight the importance of making age-specific decisions on the appropriate source-detector separation distances for the target age group [25]. NIRS studies conventionally use 20mm to 30mm separation distances for infants [55, 56] and 30mm to 35mm for children (e.g., [57, 58]) and adults [59]. Young infants and adults displayed comparable DOT sensitivity profiles at 20mm-to-30mm separation distance. However, Fu and Richards (under review) [27] showed that young infants had closer scalp-to-cortex distances than adults (3 months: 8mm; 20–24 years: 13mm). Hence, young infants may have greater DOT sensitivity to the cortex than adults (cf. [10]). The age difference was more discernible at larger separation distances. Increasing separation distances to 35mm or greater may cause more concern in terms of reduction of signal strength and increased variance with relatively less gain in improving sensitivity to the cortex for older infants compared to adults.

Detailed descriptions of changes in DOT sensitivity profiles by separation distances across developmental and adult samples can better inform age-specific designs of channel/optode placement. Several studies have used the S-D Channel DOT fluence measure to estimate the mapping between channel locations and ROIs [26–31]. For example, Zimeo Morais et al. [26] provided a toolbox that displays the channels that measure the user-specified ROI with a specificity value (1%-100%) for each channel. For the selected candidate channels, the users can also obtain their specificity to measure other (non-selected) brain regions. The channel(s) that show good discrimination to the user-selected ROI (i.e., high specificity to the ROI and low specificity to other brain and non-brain regions) are desirable. However, the toolbox uses a pool of 130 10–10 channels with a small range of separation distances for the adult head model (median = 3.60cm, lower quartile = 3.13cm, upper quartile = 3.98cm; [26]). The toolbox and existing look-up tables do not consider that the separation distance for a given channel changes with age as head sizes get bigger. Additionally, the current findings indicate that the

separation-related changes in sensitivity profile also vary by age groups. Increasing source-detector separation distances beyond 30mm for infants may reduce the channel-to-ROI discrimination. This is problematic especially when researchers use ROIs from micro-structure atlases and are interested in comparing results between age groups. Future tools for designing channel placements should allow users to optimize the constellation of optode locations based on ROIs, age group, and separation distances.

The age-related differences in sensitivity profiles reinforce the importance of using age-appropriate head models for DOT image reconstruction. The sensitivity profile estimated from Monte Carlo photon migration simulations can be recorded into a forward matrix. This is then used for solving the inverse problem in DOT image reconstruction that localizes the ROIs that showed hemoglobin concentration changes [5, 6]. Contemporary toolboxes for DOT image source reconstruction (AtlasViewer: [60]; TOAST++: [12]) use adult head models as the default option. The present study provides data that show the importance of using age-appropriate templates as an anatomical prior to improve accuracy in channel location co-registration and forward model estimation. Delgado Reyes et al. [61] and Wijeakumar et al. [62] show examples of this. They selected head models with atlas parcellations from the Neurodevelopmental MRI database [34–37] that were closely matched in age of the participants (4-month- to 2-year-olds: [61]; 4- to 48-month-olds: [62]). The head models were used for estimating the forward model and image reconstruction using the AtlasViewer [60]. The resultant voxel-wise hemodynamic concentration estimates were then entered in group-level analyses.

## Limitations

Different optical properties might need to be applied to infant and adult head models for photon migration simulations [10, 19]. There is a paucity of reference for appropriate optical property values for different tissue types in adult and infant realistic head models, given existing studies performed less fined-grained segmentation of tissue layers (scalp, skull, CSF, GM, and WM). It is also computationally challenging to systematically quantify how variations in optical properties of different media types may impact S-D Channel DOT sensitivity estimations and whether the impact may differ between infant and adult realistic head models. In addition, the present study did not differentiate optical property values for some extracerebral tissue layers including the skin, dura, muscle, and nasal cavity. The impact of optical property settings for extracerebral tissues on photon migration might differ between infant and adult head models due to the age-related differences in the thickness of these tissue media [10, 19, 27, 54]. However, Brigadoi and Cooper [19]'s validation analyses showed that variations in the CSF scattering coefficients in the infant head model and scalp absorption coefficients in the adult head model within the range of commonly adopted values did not considerably impact the DOT sensitivity estimations. Future validation studies are needed to ensure that the age differences in DOT sensitivity profiles are over and above the age-related variations in optical properties of extracerebral and brain tissue layers.

The current study did not compute DOT sensitivity by tissue types separately. Existing evidence indicated that the change of relative sensitivity as a function of source-detector separation distance displayed different shapes for GM, WM, CSF, and extracerebral tissues (scalp and skull) in infants and adults [3, 10, 18, 20]. Our data could be used to evaluate the fluence strength to the cortex based on the S-D Channel DOT scalp-to-cortex distance and the sensitivity profiles. We also could directly compare the relative sensitive to the brain versus extracerebral tissues at different separation distances for infants and adults [3, 19]. Future analyses that compare sensitivity profiles by tissue types could more precisely inform age-related differences in the optimal separation distances for sampling GM signal changes.

## Conclusions

The current study examined S-D Channel DOT sensitivity profiles in infants, children, and adults. The profiles changed systematically as a function of source-detector separation distance for all age groups. There were also important age-related differences in the S-D Channel DOT sensitivity profiles. These are particularly discernible between younger (2 weeks to 4.5 months) and older infants at larger separation distances. Our findings of age-related differences in S-D Channel DOT sensitivity profiles are useful for determining age-appropriate source-detector separation distances and placement that match sensitivity, signal strength, and penetration depth for different ages. Our findings imply that the optimization of channel configurations and accurate anatomical interpretations of NIRS/fNIRS data are dependent on developmentally sensitive estimations of S-D Channel DOT sensitivity that account for the head and cortical development. The present study has demonstrated that age-appropriate realistic head models can be used with photon migration simulations to provide anatomical guidance for standalone DOT data.

## Supporting information

**S1 File.**
(DOCX)

**S1 Table. References of optical properties used in published studies.**
(XLSX)

**S2 Table. Source-detector separation information for "all optode pairs" constructed from 10–5 optode locations.** It provides the minimum and maximum distances specified for each target separation distance (10mm to 60mm in 5mm increment) for all age groups.
(XLSX)

## Acknowledgments

The authors would like to thank researchers in the Autism Brain Imaging Data Exchange (ABIDE), Baby Connectome Project (BCP), Early Brain Development Study (EBDS), Infant Brain Imaging Study (IBIS), McCausland Center for Brain Imaging, and Pediatric Imaging, Neurocognition, and Genetics Data Repository (PING) for making the neuroimaging data available.

## Author Contributions

**Conceptualization:** Xiaoxue Fu, John E. Richards.

**Data curation:** Xiaoxue Fu, John E. Richards.

**Formal analysis:** Xiaoxue Fu.

**Funding acquisition:** John E. Richards.

**Methodology:** John E. Richards.

**Resources:** John E. Richards.

**Supervision:** John E. Richards.

**Visualization:** Xiaoxue Fu.

**Writing – original draft:** Xiaoxue Fu.

**Writing – review & editing:** Xiaoxue Fu, John E. Richards.

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
