## [Decision Letter · Decision Letter 0]

22 Feb 2021

PONE-D-20-39767

Age-related Changes in Diffuse Optical Tomography Sensitivity by Source-Detector Separation in Infancy

PLOS ONE

Dear Dr. Fu,

Thank you for submitting your manuscript to PLOS ONE. After careful consideration, we feel that it has merit but does not fully meet PLOS ONE’s publication criteria as it currently stands. Therefore, we invite you to submit a revised version of the manuscript that addresses the points raised during the review process.

We look forward to receiving your revised manuscript.

Kind regards,

Andrea Farina

Academic Editor

PLOS ONE

Journal Requirements:

Reviewers' comments:

Reviewer's Responses to Questions

**Comments to the Author**

1. Is the manuscript technically sound, and do the data support the conclusions?

Reviewer #1: Partly

Reviewer #2: Yes

2. Has the statistical analysis been performed appropriately and rigorously? 

Reviewer #1: N/A

Reviewer #2: N/A

3. Have the authors made all data underlying the findings in their manuscript fully available?

Reviewer #1: Yes

Reviewer #2: Yes

4. Is the manuscript presented in an intelligible fashion and written in standard English?

Reviewer #1: Yes

Reviewer #2: Yes

5. Review Comments to the Author

Reviewer #1: The work reported in interesting and there is potential for novel findings. However, the work in the current form requires some modifications and improvements. More specifically:

- Sentence in Introduction “DOT does not provide anatomical information about the location of the hemodynamic signal.” is unclear. DOT can be used to provide spatial information where the contrast is generated by light absorption and scattering. What would be the anatomical information of the location that the authors mean?

- Manuscript lacks an explanation on Monte Carlo simulation. Basic principles and parameters should be explained.

- Three different experimental systems and datatypes are used in DOT: time-domain, frequency-domain and continuous-wave. It seems that the authors are considering continuous-wave simulations. However, that is not mentioned.

- In the manuscript, acronym FEM is used for finite element model. However, FEM typically refers to finite element method that is a numerical method for solving partial differential equations. Further, when the authors refer to their discretisation, it should referred as finite element (FE) mesh. Not FEM mesh, if finite element method is not used.

- The authors form tetrahedral discretisation from segmented MR images. However, they use voxel-based MCX to simulate light propagation. This is inconsistent. What discretization was used in simulations? Why use two different meshes when one could also use tetrahedral mesh in Monte Carlo? If two different meshes were used, how was the interpolation between the meshes implemented?

- The authors use work “electrodes” to describe their light sources and detectors. Electrodes are not used in optical measurements. One can call those “optodes”, or just sources and detectors.

- The results show a large number of graphs. However, no statistical analysis of the results is presented. One would be keen to get some more insight or conclusions of the work. If not statistical, then maybe more spatial information. After all, Monte Carlo provides fluence in the whole volume, not only the measurement signal.

Reviewer #2: This manuscript studied the fluence change during infancy by using accurate head models with small age intervals and Monte Carlo simulations. This research topic is valuable for current DOT and fNIRS communities since the brain anatomy changes drastically from infancy to adulthood, which will have a significant impact on the fluence distribution in the human brain. This manuscript can be accepted if following issues can be properly addressed.

Major:

1. It is believed that the focus of this manuscript is how the age will impact the fluence distribution in the human brain and the importance to have age-appropriate realistic head models. However, the manuscript spends too much time discussing other topics such as separation distance. The mixed discussions on age and separation makes readers confused about the message this manuscript wants to convey. The manuscript should be organized in a way the emphasis can be easily caught by the readers. If the message is the necessity of using accurate infant brain models, the accuracy of the brain models should be discussed.

2. The terminologies used in the manuscript are inconsistent and the definitions of these terminologies are problematic. There is a mixed usage between PPL, penetration depth, separation distance, electrode, optode, etc. The author should stick with one naming convention throughout the manuscript. Also, if possible, the formula should be given to define the terminologies.

3. It is very difficult to extract information by reading the figures. The focus of this paper is about the impact of age on the fluence. However, many figures are plotting the relationship between PPL and fluence. And it is not clear why this information is important to the DOT community since earlier studies have similar analyses. The figures should be arranged in a way showing why it is important to use realistic brain models for infancy. In such a case, age should be the x-axis.

Minor:

1. The title is ambiguous. What is the focus of this title? Is it age-related fluence changes or source-detector separation.

2. In the abstract, in statement ‘the peak of the sensitivity function was largest at the

smallest separation distance and decreased as source-detector distance increased’, which brain region does the sensitivity indicate? Sensitivity in skin, skull or brain?

3. In the 3rd paragraph, last sentence ‘PPL represent the penetration depths of the photon measured in the medium’. It sounds to be an inaccurate description for PPL. PPL should be the 'photon trajectory length in a specific tissue'. Can you clarify how you computed PPL in this manuscript.

4. In Fig. 1B and C, a brain model with dense and sparse densities is shown. Which mesh did you use in the manuscript? The dense or the sparse one? Why are these two meshes shown?

5. In Fig. 1A, can the author add the legend showing the mapping between color->tissue type? Also, it looks to me the dura is pretty thick and the CSF (dark orange) is not filling between brain and skull? What is the dark blue color in the figure, which appears both outside and inside of the brain?

6. In section ‘Scalp Location’, 1st paragraph, can you add citations for 10-10 and 10-5 electrode systems?

7. In section ‘Source-Detector Channels, All Electrode Pairs’, it is confusing to use electrodes here while you are placing light source or optode in the position. Please just call it source or detector and use the same names throughout the manuscript.

8. In section ‘Photon Migration Simulations’, while the author admitted the optical properties differ considerably across studies and it is likely different values are needed for adults versus development samples, the author did not give a reason or method how these optical properties are compiled in Table 2. This needs to be more clear and discussed in limitations.

9. In Table 2, the wavelength of these optical properties is not specified. In addition, citations should be added for the optical properties, showing from which papers these values are derived.

Also, the absorption of muscle seems to be low because muscle is highly absorbing. The nasal cavity, which should be air, has an absorption coefficient of 0.0101 mm^-1. What is the reason for this? Please refer to the paper:

a) Selective photobiomodulation for emotion regulation: model-based dosimetry study.

10. In section ‘DOT Sensitivity Analyses’, last sentence, there is a statement ‘This figure shows a similar monotonic decrease in fluence strength as a function of partial path length’. However, which PPL are you referring to? And PPL and penetration depth should not be the same thing. The author should stick with one naming in the manuscript.

11. In Fig. 3, the S-D Channel DOT fluence is a 3-D distribution. But In Fig. 4 and 5, it becomes a scalar that changes depending on PPL. How did you convert it to a value? Also, int Fig. 4 and 5, the S-D Channel DOT fluence does not have a unit.

12. As mentioned in the major issues, the figures from Fig. 4 to 7 are very difficult to read. It is difficult to extract effective information from these figures. These should be re-organized.

13. In section ‘The Shape of S-D Channel DOT Fluence Sensitivity Profiles’, last sentence ‘The larger HWHM location values for the youngest age group implies the DOT sensitivity to deeper cortical areas for the youngest ages.’ However, in Fig. 7B, the HWHM for age 4 to 24 years old is the highest, which conflicts with the claim that the youngest age group has DOT sensitivity in deeper cortical areas. The same statement was made in the 5th sentence in Discussion. Please clarify.

14. In Discussion, while the findings about sensitivity change and separations for infancy are given, the insights on what may cause these changes are not given. While these may not be included in this manuscript, they can be future directions.

6. PLOS authors have the option to publish the peer review history of their article (what does this mean?). If published, this will include your full peer review and any attached files.

Reviewer #1: No

Reviewer #2: No

---

## [Author Response · Author response to Decision Letter 0]

25 Mar 2021

Journal Requirements:

We have made sure that our manuscript meets PLOS ONE’s style requirements. 

We have made sure that the grant information is accurate in the “Funding Information” and “Financial Disclosure” sections. In addition, we edited the “Acknowledgement” section to remove the grant information.

3. We note that you have indicated that data from this study are available upon request. PLOS only allows data to be available upon request if there are legal or ethical restrictions on sharing data publicly. In your revised cover letter, please address the following prompts: a) If there are ethical or legal restrictions on sharing a de-identified data set, please explain them in detail (e.g., data contain potentially identifying or sensitive patient information) and who has imposed them (e.g., an ethics committee). Please also provide contact information for a data access committee, ethics committee, or other institutional body to which data requests may be sent. b) If there are no restrictions, please upload the minimal anonymized data set necessary to replicate your study findings as either Supporting Information files or to a stable, public repository and provide us with the relevant URLs, DOIs, or accession numbers. 

We do not have legal or ethical restrictions on data sharing. We have uploaded the sensitivity profile dataset and the SAS scripts for data visualization in the Open Science Framework. We have edited the “Data and Code Availability Statement” to reflect this change. 

Reviewer 1’s Comments:

1. Sentence in Introduction “DOT does not provide anatomical information about the location of the hemodynamic signal.” is unclear. DOT can be used to provide spatial information where the contrast is generated by light absorption and scattering. What would be the anatomical information of the location that the authors mean?

Thanks for pointing this out. We agree that DOT instruments with dense, multi-distance channels have improved spatial resolutions. We acknowledge this in the first paragraph: “Multi-channel NIRS instruments measures hemoglobin changes in the scalp channel space. DOT instruments use overlapping and channels with multiple separation distances to enhance spatial resolution (Custo et al., 2010)”. We also specified: “However, both types of the scalp-based measurement do not directly provide anatomical information about the brain regions of the hemodynamic signal. Localizing the channel-wise signals to specific brain regions requires comprehensive understanding of the DOT sensitivity profile.” We noted later in the paragraph: “The forward model can then guide DOT image reconstruction to recover the brain locations of hemoglobin concentration changes (Cooper et al., 2012; Culver, Siegel, Stott, & Boas, 2003). 

2. Manuscript lacks an explanation on Monte Carlo simulation. Basic principles and parameters should be explained.

We agree the Monte Carlo simulation technique should be explained in more details. We have now highlighted the basic principle in “The Use of Monte Carlo Simulations to Estimate DOT Sensitivity Profile” section in the Introduction: “A goal of DOT is to model how hemoglobin changes at the channel space (what is measured) correspond to changes in optical properties within the brain (to be determined). This requires modeling the DOT sensitivity. One approach to solve the forward model is using finite element method (FEM) quantitative algorithms to describe photon propagations through heterogeneous tissue structures such as the human head (NIRFAST: Dehghani et al., 2009; TOAST++: Schweiger & Arridge, 2014). Alternatively, Monte Carlo simulations provide numerical solutions to the radiative transport equation that models the photon migration through the head tissues based on fewer assumptions (Huppert, Franceschini, & Boas, 2009; Jacques & Wang, 1995). The Monte Carlo method is computationally demanding but provide more accurate solutions for the forward model (Wheelock, Culver, & Eggebrecht, 2019).” We have also explained in more details the input parameters and basic steps for the simulations in the same paragraph. In addition, we moved details on the inputs of the simulations used in the present study from the Supporting Information to the main text Methods section (“Photon Migration Stimulations”). 

3. Three different experimental systems and datatypes are used in DOT: time-domain, frequency-domain and continuous-wave. It seems that the authors are considering continuous-wave simulations. However, that is not mentioned.

Thank you for pointing out the important detail. Our Monte Carlo simulations are for the continuous-wave systems indeed. We have stated the information in the “The Use of Monte Carlo Simulations to Estimate DOT Sensitivity Profile” section (second paragraph) in the Introduction: “The Monte Carlo simulations commonly model fluence distributions for the continuous-wave systems as they are more widely implemented in commercial NIRS instruments than the time-domain and frequency-domain systems (Huppert et al., 2009; Wheelock et al., 2019). The continuous-wave systems measure fluence at the detector location but do not separately quantify the effects of light absorption and scattering.”

4. In the manuscript, acronym FEM is used for finite element model. However, FEM typically refers to finite element method that is a numerical method for solving partial differential equations. Further, when the authors refer to their discretisation, it should referred as finite element (FE) mesh. Not FEM mesh, if finite element method is not used.

Thank you very much for the suggestion. We have changed “FEM mesh” to “finite element (FE) mesh”. 

5. The authors form tetrahedral discretisation from segmented MR images. However, they use voxel-based MCX to simulate light propagation. This is inconsistent. What discretization was used in simulations? Why use two different meshes when one could also use tetrahedral mesh in Monte Carlo? If two different meshes were used, how was the interpolation between the meshes implemented?

We apologize for the confusion. We used dense tetrahedral discretization. We stated in the “Mesh Generation” section: “The dense “segmented FE mesh” was used for MCX (Fang & Boas, 2009) to find a segment element that is closest to an optode position. In addition, we added in the “Photon Migration Simulations” section: “The voxel-based MCX simulations do not require “segmented FE mesh” (Fang & Boas, 2009). The default voxel size 1×1×1mm3 was used.” We removed information about the “sparse meshes” from the main text and the Supporting Information. Sparse meshes were not used in the procedures implemented for the current paper. 

6. The authors use work “electrodes” to describe their light sources and detectors. Electrodes are not used in optical measurements. One can call those “optodes”, or just sources and detectors.

Thank you for pointing out the potential confusion. The 10-10 and 10-5 systems are designed for EEG electrode positions (Jurcak, Tsuzuki, & Dan, 2007). We are using “electrode(s)” when referring specifically to the 10-10 or 10-5 electrode systems. We used “optode(s)” and channel(s)” when referring specifically to the NIRS sensors and the positioning of these sensors. We have gone through the main text and Supporting Information to ensure that this rule is consistently applied. 

7. The results show a large number of graphs. However, no statistical analysis of the results is presented. One would be keen to get some more insight or conclusions of the work. If not statistical, then maybe more spatial information. After all, Monte Carlo provides fluence in the whole volume, not only the measurement signal.

We have now added summary paragraphs in the Results section to highlight the key insights that can be drawn from each finding. For the “S-D Channel DOT Sensitivity Profiles by Source-Detector Separation Distances” results, we added: “Summary: Our findings showed that the DOT sensitivity profiles show robust characteristics across age groups but also important age-related differences. In all age groups, the S-D Channel DOT fluence decreased exponentially as the light traveled deeper into the head tissues. The sensitivity profiles for the 20-to-24-year-olds showed that as the separation distances increased, the peak of the fluence value decreased, but the fluence strength declined less rapidly as the light traveled through the tissues. These patters confirmed the robust characters of DOT sensitivity profiles found in previous studies using adult head models (Mansouri, L'Huillier, Kashou, & Humeau, 2010; Strangman, Li, & Zhang, 2013; Strangman, Zhang, & Li, 2014; Wang, Ayaz, & Izzetoglu, 2019). Older infants (6 months to 2 years) displayed different DOT sensitivity profiles than young infants (2 weeks to 4.5 months) and adults, and the age-group differences are more discernible in larger source-detector separation distances. 

For “The Variance of S-D Channel DOT Sensitivity Profiles”, we summarized that “Between-channel variance was relatively stable for the child and adult. The variance increased as a function of source-detector separation distance in infant groups. The increase was most observable in older infants (6 months to 2 years).” 

For “The Shape of S-D Channel DOT Sensitivity Profiles”, we highlighted that “the shape of DOT sensitivity profiles corroborated existing evidence that the fluence distribution extends deeper into the head tissues as the source-detector distance increases (Mansouri et al., 2010; Strangman et al., 2013; Strangman et al., 2014; Wang et al., 2019). This pattern was found in all age groups. The important age-related difference is that the same proportion of fluence strength carries deeper into the head tissues for the young infant, child, and adult groups across all separation distances.”

Our current analyses have focused on presenting spatial information of the S-D Channel DOT sensitivity profile. The sensitivity profile is presented as changes in S-D Channel DOT as the function of sampling depth. We also examined changes in half width half maximum locations of the S-D Channel DOT fluence distribution as a function of source-detector separation distances. 

Reviewer 2’s Comments:

1. It is believed that the focus of this manuscript is how the age will impact the fluence distribution in the human brain and the importance to have age-appropriate realistic head models. However, the manuscript spends too much time discussing other topics such as separation distance. The mixed discussions on age and separation makes readers confused about the message this manuscript wants to convey. The manuscript should be organized in a way the emphasis can be easily caught by the readers. If the message is the necessity of using accurate infant brain models, the accuracy of the brain models should be discussed.

Thank you for your suggestion. The two main points that our findings convey are that 1) the basic characteristics of the DOT sensitivity profiles are present for all age groups, and 2) there are also age differences in DOT sensitivity profiles as a function of source-detector separation distances. In response to Reviewer 1 Comment 7, we have now added summaries of the key findings in the Results section. 

We also edited the Discussion to highlight the central messages. Specificity, in the “DOT sensitivity Profile by Source-Detector Separation Distances” section, we mirrored the organization in the Introduction. The first paragraph stated the common characteristics of the DOT sensitivity profiles across all age groups. The second paragraph highlighted the age-group differences, especially between the young infant groups (2 weeks to 4.5 months) and the older infant groups (6 months to 2 years). This paragraph commented on age-related differences in S-D channel DOT fluence as a function of sampling depth across separation distances, the variance of the sensitivity profiles, and the shape of the sensitivity profiles across separation distances. In the “Applications of Age-specific DOT Sensitivity Profiles” section of the Discussion, we argued that our findings implied that optimal design of channel configurations that maximizes DOT sensitivity to the underlying region(s) of interest (ROIs) and accurate DOT image reconstruction rely on the use of age-specific head models. This point is also underscored in the Conclusions. 

2. The terminologies used in the manuscript are inconsistent and the definitions of these terminologies are problematic. There is a mixed usage between PPL, penetration depth, separation distance, electrode, optode, etc. The author should stick with one naming convention throughout the manuscript. Also, if possible, the formula should be given to define the terminologies.

Thank you for pointing out the inconsistencies. We have changed “PPL” to “sampling depth” in response to Comment 6. We have described the computation of the sampling depth in our response. We used “penetration depth” to refer to the depth that the S-D channel DOT fluence distribution (an example shown in Figure 3) extends to the head model (e.g. Huppert et al., 2009; Mansouri et al., 2010). We agree that “penetration depth” is not equivalent to the “sampling depth” we computed. We have changed “penetration depth” to “sampling depth” when describing our findings. For example, in the second paragraph of the Results: “Compared to the sensitivity profile at shorter separation distances, the same fluence value was carried to greater sampling depth at larger separation.” 

We have now specified the “source-detector separation distance” in the “Source-Detector Channels, All Optode Pairs” section in the Materials and Methods. To summarize, we recorded the distance between each of the optode pairs. Source-detector separation distance was half of the distance between the optode pair. We have changed the phrases “source-detector distance(s)”, “source-detector separation(s)”, “distance(s)”, “separation(s)”, and “channel separation” to “source-detector separation distance(s)” or “separation distance(s)” in the main text and Supporting Information. 

In response to Reviewer 1 Comment 6, we have changed “electrode(s)” to “optode(s) and “electrode paring” to “channel” in appropriate places. In addition, we changed “depth sensitivity profile” to “S-D channel DOT sensitivity profile” to keep consistency. We defined the S-D channel DOT sensitivity profile as “S-D Channel DOT fluence as a function of the depth between channel location and the voxel sampled in the fluence distribution (“sampling depth”)” (the first paragraph of the Results). 

3. It is very difficult to extract information by reading the figures. The focus of this paper is about the impact of age on the fluence. However, many figures are plotting the relationship between PPL and fluence. And it is not clear why this information is important to the DOT community since earlier studies have similar analyses. The figures should be arranged in a way showing why it is important to use realistic brain models for infancy. In such a case, age should be the x-axis.

Thank you for the suggestion. We plotted S-D Channel DOT fluence as a function of sampling depth to ensure that the current study can replicate the findings of existing studies using adult head models. That is, the fluence value decreased monotonically and exponentially with increased depth at all separation distances (Strangman et al., 2013). In addition, the shape of the DOT sensitivity profile (S-D Channel DOT fluence as a function of sampling depth fluence) will change from short to longer separation distances. We have also highlighted in “The Present Study” section of the Introduction: “The study will provide novel evidence revealing developmental changes in how the DOT sensitivity profiles vary across separation distances.”

All the current figures present visualization of age-related differences. Separate lines were used to display age group as a categorical variable. However, we have taken the reviewer’s suggestion to highlight age-group differences in our visualizations. These additional figures are presented in the Supporting Information. For example, Figure 4A, 4B, and 4C present the S-D Channel DOT sensitivity profile at 20mm, 30mm, and 50mm separation distance, respectively. Age groups were plotted as individual bars. The figures help to visualize age-group differences in the fluence values at each sampling depth interval. Additional figures were also produced for the variance of the S-D Channel DOT fluence (Supporting Information Figure 5) and the half width half maximum (HWHM) locations (Supporting Information Figure 7). We have referenced these additional figures in the Main Text. In response to Reviewer 1 Comment 7, we have also summarized the key information obtained from the visualizations after each section of the Results. We have chosen to keep our original figures in the Main Text, but we will be willing to move additional figures from the Supporting Information to the Main Text based on the reviewer’s suggestion. 

4. The title is ambiguous. What is the focus of this title? Is it age-related fluence changes or source-detector separation.

We have now changed the title to “Age-related changes in diffuse optical tomography sensitivity profiles in infancy. This is to emphasize the age-related fluence changes. 

5. In the abstract, in statement ‘the peak of the sensitivity function was largest at the smallest separation distance and decreased as source-detector distance increased’, which brain region does the sensitivity indicate? Sensitivity in skin, skull or brain?’

Thank you for pointing out the lack of specification. We did not assess DOT sensitivity profiles in each of the tissue types separately as done in published studies (e.g. Strangman, Li, Zhang, 2013; Strangman, Zhang, Li, 2014). Hence, we cannot make a specific statement on whether the sensitivity profiles are for the skin, skull, or brain. We have acknowledged the limitation in the Discussion (“Limitation” section). We now have added a specification in the sentence: “The peak of the sensitivity function in the head was largest at the smallest separation distance and decreased as source-detector distance increased.”

6. In the 3rd paragraph, last sentence ‘PPL represent the penetration depths of the photon measured in the medium’. It sounds to be an inaccurate description for PPL. PPL should be the 'photon trajectory length in a specific tissue'. Can you clarify how you computed PPL in this manuscript.

Thank you for raising the issue. We agree that the construct computed in the current paper does not represent “photon trajectory length in a specific tissue”, as defined in the literature we cited (Mansouri et al., 2010; Strangman et al, 2014; Wang et al, 2019). We defined the construct in the “Depth Sensitivity Profile by Source-Detector Separation Distances” section in the Materials and Methods. It is the distance (depth) from the channel location to the voxel with the S-D channel DOT fluence estimation. We computed this construct to display the changes in fluence as photons traveled though the head tissues by separation distances and age groups (as displayed in Figure 4 and 5). We are referring this as “sampling depth”, in order to provide an abbreviated term for the figures and for ease of descriptions. We have stated the abbreviated name in the “S-D Channel DOT Sensitivity Profile by Source-Detector Separation Distances” section. In the same section, we provided more details of the computations: “For each participant and each channel with the target separation distance (e.g. 400 channels with 10mm separation distance), we computed the S-D channel DOT fluence value at each sampling depth. The fluence values were then averaged across channels by sampling depth (1 to 100), target separation distance, and participant. The fluence values were then averaged by separation distance and age group”. Figure 5B, 5C, and 5D displays the averaged fluence values as a function of the sampling depth for the 20mm, 30mm, 50mm target separation distance by age groups. 

We have removed “PPL” from the third paragraph. We changed the sentences to: “DOT sensitivity is quantified as each location at which a photon traverses through a given tissue region, or “sampling depth”. It represents the depth of the photon measured in the head medium.”. We have changed “PPL” to “sampling depth” in the rest of the manuscript, Figure 4, 5, and 6, as well as Supporting Information. 

7. In Fig. 1B and C, a brain model with dense and sparse densities is shown. Which mesh did you use in the manuscript? The dense or the sparse one? Why are these two meshes shown?

We apologize for the confusion. As in our response to Comment #5 from the Reviewer 1, we used the dense “segmented FE mesh” in the current study to locate points on the scalp that were closest to the optode positions. We have removed the sparse mesh in Figure 1 and all related information from the Main Text and the Supporting Information.

8. In Fig. 1A, can the author add the legend showing the mapping between color->tissue type? Also, it looks to me the dura is pretty thick and the CSF (dark orange) is not filling between brain and skull? What is the dark blue color in the figure, which appears both outside and inside of the brain?

Thank you for the suggestion, and we apologize for the confusion. We have now added a legend and edited the caption for Figure 1 A. The dura is the darker blue ribbon underneath the skull (lighter blue). The CSF (orange) is inside of the dura. The dark blue is the gray matter and the purple is the white matter. 

9. In section ‘Scalp Location’, 1st paragraph, can you add citations for 10-10 and 10-5 electrode systems?

We have added a citation for the 10-10 and 10-5 systems: “The locations for the 10-10 and 10-5 systems (Jurcak, Tsuzuki, & Dan, 2007) were constructed on each head MRI volume. 

10. In section ‘Source-Detector Channels, All Electrode Pairs’, it is confusing to use electrodes here while you are placing light source or optode in the position. Please just call it source or detector and use the same names throughout the manuscript.

Thank you for pointing out the confusion. We have changed the “all electrode pairs” to “all optode pairs”. We have changed “electrode(s)” to “optode(s) as appropriate in response to Reviewer 1 and Reviewer 2’s comments.

11. In section ‘Photon Migration Simulations’, while the author admitted the optical properties differ considerably across studies and it is likely different values are needed for adults versus development samples, the author did not give a reason or method how these optical properties are compiled in Table 2. This needs to be more clear and discussed in limitations.

Thank you for the suggestion. We have now provided references and reasons for our selection of the optical property for each tissue type in Table 2. We have now discussed the potential issues with using the same optical property settings across age groups in the first paragraph of “Limitations”. We pointed out that “the impact of optical property settings for extracerebral tissues on photon migration might differ between infant and adult head models due to the age-related differences in the thickness of these tissue media (Brigadoi & Cooper, 2015; Dehaes et al., 2011; Fu & Richards, under review; Fukui, Ajichi, & Okada, 2003).” We cited Brigadoi and Cooper (2015)’s validation results indicating that variations in scalp and CSF optical properties within the common ranges might not have substantial impacts on DOT sensitivity estimations using adult and infant head models. However, we argued that “future validation studies are needed to ensure that the age differences in DOT sensitivity profiles are over and above the age-related variations in optical properties of extracerebral and brain tissue layers”.

12. In Table 2, the wavelength of these optical properties is not specified. In addition, citations should be added for the optical properties, showing from which papers these values are derived. Also, the absorption of muscle seems to be low because muscle is highly absorbing. The nasal cavity, which should be air, has an absorption coefficient of 0.0101 mm^-1. What is the reason for this? Please refer to the paper: Selective photobiomodulation for emotion regulation: model-based dosimetry study.

Thank you for providing reference. The wavelength of the optical properties in the current paper is 690nm. This information is specified in the legend of Table 2. We have now provided the citations and reasons for the optical property selected for each tissue type in Table 2. We did differentiate the different non-brain tissue types: skin, dura, muscle, and nasal cavity. We agree with the reviewer that different optical properties should have been set for skin, muscle, and nasal cavity (as in Cassano et al., 2019). We have acknowledged this limitation in the Discussion. Our optical property input for the nasal cavity could also be modified. However, we expect the effect of the setting to the fluence estimation is minimal, as the photon is unlikely to reach to the nasal cavity before exiting the head media or being absorbed by the media. 

13. In section ‘DOT Sensitivity Analyses’, last sentence, there is a statement ‘This figure shows a similar monotonic decrease in fluence strength as a function of partial path length’. However, which PPL are you referring to? And PPL and penetration depth should not be the same thing. The author should stick with one naming in the manuscript.

Thank you for pointing out the issue with the use of terminology. We have changed the naming “PPL” to “sampling depth”. Please refer to our response for Comment 6 for more detailed explanations. We described the computation of sampling depth in the following section “Depth Sensitivity profile by Source-Detector Separation Distances”. In the description of Figure 3, we changed the sentence to: “Figure 3 shows the S-D Channel DOT fluence plotted on an MRI for a single source-detector channel. This figure shows a similar monotonic decrease in fluence strength as photons traveled deeper into the head tissues.”

We agree that “penetration depth” and “sampling depth” are not the same construct. We have changed “penetration depth” to “sampling depth” when describing our findings (please also see our response to Comment 2.

14. In Fig. 3, the S-D Channel DOT fluence is a 3-D distribution. But In Fig. 4 and 5, it becomes a scalar that changes depending on PPL. How did you convert it to a value? Also, int Fig. 4 and 5, the S-D Channel DOT fluence does not have a unit.

We have clarified in the “DOT Sensitivity Analyses” section of the Materials and Methods: “(S-D Channel DOT fluence)..is a unitless measure that represents the sensitivity of the DOT measure for detecting changes in optical properties at a given point inside the head medium (PMDF; Brigadoi, Aljabar, Kuklisova-Murgasova, Arridge, & Cooper, 2014; Brigadoi & Cooper, 2015; or "3-point Green's function"; Strangman et al., 2013). Figure 3 shows displays the S-D channel DOT fluence distribution at a channel location for all voxels in the head model space. In the next paragraph, we have now specified: “The S-D Channel DOT fluence was extracted voxel by voxel. We recorded the fluence value and the distance (depth) from the channel location to the voxel with the fluence estimation. We refer the distance from the channel to the location of the fluence value inside the head model as “sampling depth” for short. The S-D channel DOT sensitivity profile was defined as the fluence value at the voxel as a function of the sampling depth.” Figure 4 and 5 displayed the S-D channel DOT sensitivity profiles across source-detector separation distances. 

15. As mentioned in the major issues, the figures from Fig. 4 to 7 are very difficult to read. It is difficult to extract effective information from these figures. These should be re-organized.

We have now added more detailed descriptions of the variables in the figure captions. As our response to Reviewer 1, we have now added summaries of the key findings displayed in Figure 4 to Figure 7 in the Results section. We have also explained our rationales for these visualizations in our response to Comment 3. 

16. In section ‘The Shape of S-D Channel DOT Fluence Sensitivity Profiles’, last sentence ‘The larger HWHM location values for the youngest age group implies the DOT sensitivity to deeper cortical areas for the youngest ages.’ However, in Fig. 7B, the HWHM for age 4 to 24 years old is the highest, which conflicts with the claim that the youngest age group has DOT sensitivity in deeper cortical areas. The same statement was made in the 5th sentence in Discussion. Please clarify.

We apologize for the confusion. We have now stated in the “Shape of S-D Channel DOT Sensitivity Profile”: “The mean HWHM locations across all separation distances were greater in the youngest infants (2-week through 4.5 month old age groups, Figure 7A and 7B solid line, aqua color), children and adults (Figure 7A and 7B green dashed line)”. We also stated that “The larger HWHM location values imply the DOT sensitivity to deeper cortical areas for the youngest infants, children, and adults.” In the first paragraph of the Discussion, we added: “The shape of S-D channel DOT sensitivity profile was similar between the younger infant groups and child and adult groups.” We placed greater emphasis on discussing the differences in S-D Channel DOT sensitivity profiles across separation distances among infant groups. The child and adult groups were used as reference groups to show broad differences between infants and older ages. 

17. In Discussion, while the findings about sensitivity change and separations for infancy are given, the insights on what may cause these changes are not given. While these may not be included in this manuscript, they can be future directions.

Thank you for the suggestion. We have now discussed the possibility that “developmental changes in the morphology of the head and the brain may contribute to the age-related differences in S-D channel DOT sensitivity profiles” (the third paragraph of the “S-D Channel DOT Sensitivity Profile by Source-Detector Separation Distances” section in the Discussion. We acknowledged in the end of the paragraph that “a direction for future research is to characterize morphological changes in specific tissue layers that are linked to age-related differences in S-D channel DOT sensitivity profiles across separation distances”.

---

## [Decision Letter · Decision Letter 1]

10 May 2021

Age-related Changes in Diffuse Optical Tomography Sensitivity Profiles in Infancy

PONE-D-20-39767R1

Dear Dr. Fu,

We’re pleased to inform you that your manuscript has been judged scientifically suitable for publication and will be formally accepted for publication once it meets all outstanding technical requirements.

Kind regards,

Andrea Farina

Academic Editor

PLOS ONE

Additional Editor Comments (optional):

I kindly ask you to edit Fig.3 specifying a colorbar and the source-detector separation related to the simulation.

Reviewers' comments:

Reviewer's Responses to Questions

**Comments to the Author**

1. If the authors have adequately addressed your comments raised in a previous round of review and you feel that this manuscript is now acceptable for publication, you may indicate that here to bypass the “Comments to the Author” section, enter your conflict of interest statement in the “Confidential to Editor” section, and submit your "Accept" recommendation.

Reviewer #1: All comments have been addressed

Reviewer #2: All comments have been addressed

2. Is the manuscript technically sound, and do the data support the conclusions?

Reviewer #1: Yes

Reviewer #2: Yes

3. Has the statistical analysis been performed appropriately and rigorously? 

Reviewer #1: Yes

Reviewer #2: Yes

4. Have the authors made all data underlying the findings in their manuscript fully available?

Reviewer #1: Yes

Reviewer #2: Yes

5. Is the manuscript presented in an intelligible fashion and written in standard English?

Reviewer #1: Yes

Reviewer #2: Yes

6. Review Comments to the Author

Reviewer #1: The authors have addressed all comments by reviewers thoroughly, and they have improved the manuscript clearly. The manuscript now provides a very nice and well written description of the research. I recommend the manuscript to be accepted for publication in its current form.

Reviewer #2: Age-related sensitivity profile in infancy is a very valuable topic for current fNIRS study. All my earlier comments have been addressed in detail. I recommend to accept this manuscript for the journal.

7. PLOS authors have the option to publish the peer review history of their article (what does this mean?). If published, this will include your full peer review and any attached files.

Reviewer #1: No

Reviewer #2: No

---

## [Editor Report · Acceptance letter]

31 May 2021

PONE-D-20-39767R1 

Age-related Changes in Diffuse Optical Tomography Sensitivity Profiles in Infancy 

Dear Dr. Fu:

I'm pleased to inform you that your manuscript has been deemed suitable for publication in PLOS ONE. Congratulations! Your manuscript is now with our production department. 

Kind regards, 

on behalf of

Dr. Andrea Farina 

Academic Editor

PLOS ONE